# Indicative Summarization of Long Discussions

**Shahbaz Syed** [†*]    **Dominik Schwabe** [†*]    **Khalid Al-Khatib** [‡]    **Martin Potthast** [†§]

[†]Leipzig University        [‡]University of Groningen        [§]ScaDS.AI

shahbaz.syed@uni-leipzig.de

## Abstract

Online forums encourage the exchange and discussion of different stances on many topics. Not only do they provide an opportunity to present one's own arguments, but may also gather a broad cross-section of others' arguments. However, the resulting long discussions are difficult to overview. This paper presents a novel unsupervised approach using large language models (LLMs) to generating indicative summaries for long discussions that basically serve as tables of contents. Our approach first clusters argument sentences, generates cluster labels as abstractive summaries, and classifies the generated cluster labels into argumentation frames resulting in a two-level summary. Based on an extensively optimized prompt engineering approach, we evaluate 19 LLMs for generative cluster labeling and frame classification. To evaluate the usefulness of our indicative summaries, we conduct a purpose-driven user study via a new visual interface called DISCUSSION EXPLORER: It shows that our proposed indicative summaries serve as a convenient navigation tool to explore long discussions.[1]

## 1 Introduction

Online discussion forums are a popular medium for discussing a wide range of topics. As the size of a community grows, so does the average length of the discussions held there, especially when current controversial topics are discussed. On ChangeMyView (CMV),[2] for example, discussions often go into the hundreds of arguments covering many perspectives on the topics in question. Initiating, participating in, or reading discussions generally has two goals: to learn more about others' views on a topic and/or to share one's own.

To help their users navigate large volumes of arguments in long discussions, many forums offer basic features to sort them, for example, by time of creation or popularity. However, these alternative views may not capture the full range of perspectives

exchanged, so it is still necessary to read most of them for a comprehensive overview. In this paper, we depart from previous approaches to summarizing long discussions by using *indicative* summaries instead of *informative* summaries.[3] Figure 1 illustrates our three-step approach: first, the sentences of the arguments are clustered according to their latent subtopics. Then, a large language model generates a concise abstractive summary for each cluster as its label. Finally, the argument frame (Chong and Druckman, 2007; Boydstun et al., 2014) of each cluster label is predicted as a generalizable operationalization of perspectives on a discussion's topic. From this, a hierarchical summary is created in the style of a table of contents, where frames act as headings and cluster labels as subheadings. To our knowledge, indicative summaries of this type have not been explored before (see Section 2).

Our four main contributions are: (1) A fully unsupervised approach to indicative summarization of long discussions (Section 3). We develop robust prompts for generative cluster labeling and frame assignment based on extensive empirical evaluation and best practices (Section 4). (2) A comprehensive evaluation of 19 state-of-the-art, prompt-based, large language models (LLMs) for both tasks, supported by quantitative and qualitative assessments (Section 5). (3) A user study of the usefulness of indicative summaries for exploring long discussions (Section 5). (4) DISCUSSION EXPLORER, an interactive visual interface for exploring the indicative summaries generated by our approach and the corresponding discussions.[4] Our results show that the GPT variants of OpenAI (GPT3.5, ChatGPT, and GPT4) outperform all other open source models at the time of writing. LLaMA and T0 perform well, but are not competitive with the GPT models. Regarding the usefulness of the summaries, users preferred our summaries to alternative views to explore long discussions with hundreds of arguments.

---

*Equal contribution.

[1]Code: https://github.com/webis-de/EMNLP-23

[2]https://www.reddit.com/r/changemyview/

[3]Unlike an informative summary, an indicative summary does not capture as much information as possible from a text, but only its gist. This makes them particularly suitable for long documents like books in the form of tables of contents.

[4]https://discussion-explorer.web.webis.de/

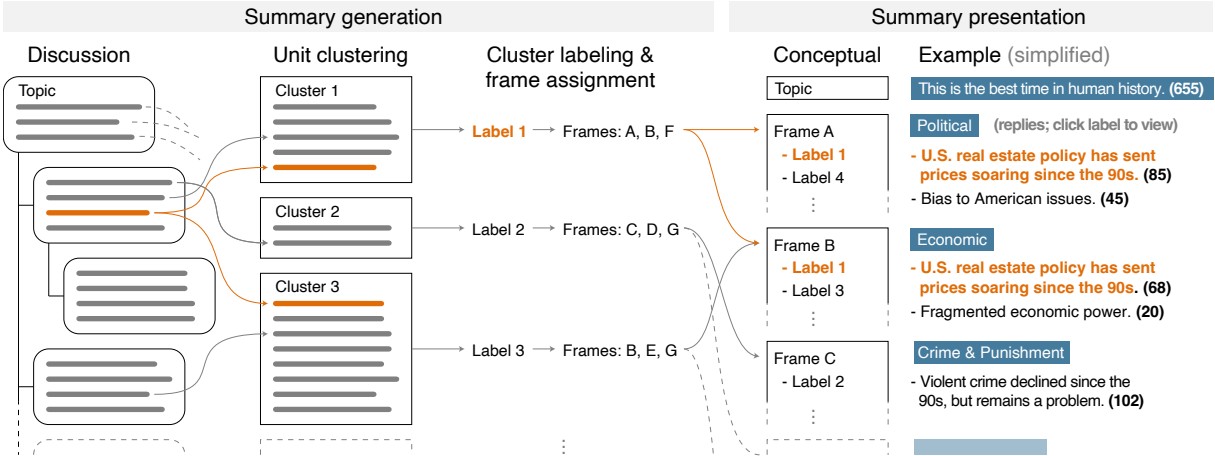

Figure 1: Left: Illustration of our approach to generating indicative summaries for long discussions. The main steps are (1) unit clustering, (2) generative cluster labeling, and (3) multi-label frame assignment in order of relevance. Right: Conceptual and exemplary presentation of our indicative summary in table of contents style. Frames act as headings and the corresponding cluster labels as subheadings.

## 2 Related Work

Previous approaches to generating discussion summaries have mainly focused on generating extractive summaries, using two main strategies: extracting significant units (e.g., responses, paragraphs, or sentences), or grouping them into specific categories, which are then summarized. In this section, we review the relevant literature.

### 2.1 Extractive Summarization

Extractive approaches use supervised learning or domain-specific heuristics to extract important entities from discussions as extractive summaries. For example, Klaas (2005) summarized UseNet newsgroup threads by considering thread structure and lexical features to measure message importance. Tigelaar et al. (2010) identified key sentences based on author names and citations, focusing on coherence and coverage in summaries. Ren et al. (2011) developed a hierarchical Bayesian model for tracking topics, using a random walk algorithm to select representative sentences. Ranade et al. (2013) extracted topic-relevant and emotive sentences, while Bhatia et al. (2014) and Tarnpradab et al. (2017) used dialogue acts to summarize question-answering forum discussions. Egan et al. (2016) extracted key points using dependency parse graphs, and Kano et al. (2018) summarized Reddit discussions using local and global context features. These approaches generate informative summaries, substituting discussions without back-referencing to them.

### 2.2 Grouping-based Summarization

Grouping-based approaches group discussion units like posts or sentences, either implicitly or explicitly. The groups are based on queries, aspects, topics, dialogue acts, argument facets, or key points annotated by experts. Once the units are grouped, individual summaries are generated for each group by selecting representative members, respectively.

This *grouping-then-summarization* paradigm has been primarily applied to multi-document summarization of news articles (Radev et al., 2004). Follow-up work proposed cluster link analysis (Wan and Yang, 2008), cluster sentence ranking (Cai et al., 2010), and density peak identification in clusters (Zhang et al., 2015). For abstractive multi-document summarization, Nayeem et al. (2018) clustered sentence embeddings using a hierarchical agglomerative algorithm, identifying representative sentences from each cluster using TextRank (Mihalcea and Tarau, 2004) on the induced sentence graph. Similarly, Fuad et al. (2019) clustered sentence embeddings and selected subsets of clusters based on importance, coverage, and variety. These subsets are then input to a transformer model trained on the CNN/DailyMail dataset (Nallapati et al., 2016) to generate a summary. Recently, Ernst et al. (2022) used agglomerative clustering of salient statements to summarize sets of news articles, involving a supervised ranking of clusters by importance.

For Wikipedia discussions, Zhang et al. (2017) proposed the creation of a dynamic summary tree to ease subtopic navigation at different levels of detail, requiring editors to manually summarize each

tree node's cluster. Misra et al. (2015) used summarization to identify arguments with similar aspects in dialogues from the Internet Argument Corpus (Walker et al., 2012). Similarly, Reimers et al. (2019) used agglomerative clustering of contextual embeddings and aspects to group sentence-level arguments. Bar-Haim et al. (2020a,b) examined the mapping of debate arguments to key points written by experts to serve as summaries.

Our approach clusters discussion units, but instead of a supervised selection of key cluster members, we use vanilla LLMs for abstractive summarization. Moreover, our summaries are hierarchical, using issue-generic frames as headings (Chong and Druckman, 2007; Boydstun et al., 2014) and generating concise abstractive summaries of corresponding clusters as subheadings. Thus our approach is unsupervised, facilitating a scalable and generalizable summarization of discussions.

## 2.3 Cluster Labeling

Cluster labeling involves assigning representative labels to document clusters to facilitate clustering exploration. Labeling approaches include comparing term distributions (Manning et al., 2008), selecting key terms closest to the cluster centroid (Role and Nadif, 2014), formulating key queries (Gollub et al., 2016), identify keywords through hypernym relationships (Poostchi and Piccardi, 2018), and weak supervision to generate topic labels Popa and Rebedea (2021). These approaches often select a small set of terms as labels that do not describe a cluster's contents in closed form. Our approach overcomes this limitation by treating cluster labeling as a zero-shot abstractive summarization task.

## 2.4 Frame Assignment

Framing involves emphasizing certain aspects of a topic for various purposes, such as persuasion (Entman, 1993; Chong and Druckman, 2007). Frame analysis for discussions provides insights into different perspectives on a topic (Morstatter et al., 2018; Liu et al., 2019). It also helps to identify biases in discussions resulting, e.g., from word choice (Hamborg et al., 2019b,a). Thus, frames can serve as valuable reference points for organizing long discussions. We use a predefined inventory of media frames (Boydstun et al., 2014) for discussion summarization. Instead of supervised frame assignment (Naderi and Hirst, 2017; Ajjour et al., 2019; Heinisch and Cimiano, 2021), we use prompt-based LLMs for more flexibility.

# 3 Indicative Discussion Summarization

Our indicative summarization approach takes the sentences of a discussion as input and generates a summary in the form of a table of contents, as shown in Figure 1. Its three steps consist of clustering discussion sentences, cluster labeling, and frame assignment to cluster labels.

## 3.1 Unit Clustering

Given a discussion, we extract its sentences as discussion units. The set of sentences is then clustered using the density-based hierarchical clustering algorithm HDBSCAN (Campello et al., 2013). Each sentence is embedded using SBERT (Reimers and Gurevych, 2019) and these embeddings are then mapped to a lower dimensionality using UMAP (McInnes et al., 2017).[5] Unlike previous approaches that rank and filter clusters to generate informative summaries (Ernst et al., 2022; Syed et al., 2023), our summaries incorporate all clusters. The sentences of each cluster are ranked by centrality, which is determined by the $\lambda$ value of HDBSCAN. A number of central sentences per cluster are selected as input for cluster labeling by abstractive summarization.

**Meta-sentence filtering** Some sentences in a discussion do not contribute directly to the topic, but reflect the interaction between its participants. Examples include sentences such as "I agree with you." or "You are setting up a straw man." Pilot experiments have shown that such meta-sentences may cause our summarization approach to include them in the final summary. As these are irrelevant to our goal, we apply a corpus-specific and channel-specific meta-sentence filtering approach, respectively. Corpus-specific filtering is based on a small set of frequently used meta-sentences $M$ in a large corpus (e.g., on Reddit). It is bootstrapped during preprocessing, and all sentences in it are omitted by default.[6]

Our pilot experiments revealed that some sentences in discussions are also channel-specific (e.g., for the ChangeMyView Subreddit). Therefore, we augment our sentence clustering approach by adding a random sample $M' \subset M$ to the set of sentences $D$ of each individual discussion before clustering, where $|M'| = \max\{300, |D|\}$. The maximum value for the number of meta-sentences $|M'|$

---

[5]Implementation details are given in Appendix B.
[6]The set is used like a typical stop word list, only for sentences.

is chosen empirically, to maximize the likelihood that channel-specific meta-sentences are clustered with corpus-specific ones. After clustering the joint set of meta-sentences and discussion sentences $D \cup M'$, we obtain the clustering $\mathcal{C}$. Let $m_C = |C \cap M'|$ denote the number of meta-sentences and $d_C = |C \cap D|$ the number of discussion sentences in a cluster $C \in \mathcal{C}$. The proportion of meta-sentences in a cluster is then estimated as $P(M'|C) = \frac{m_C}{m_C + d_C}$.

A cluster $C$ is classified as a meta-sentence cluster if $P(M'|C) > \theta \cdot P(M')$, where $P(M') = \frac{|M'|}{|D|}$ assumes that meta-sentences are independent of others in a discussion. The noise threshold $\theta = \frac{2}{3}$ was chosen empirically. Sentences in a discussion that either belong to a meta-sentence cluster or whose nearest cluster is considered to be one are omitted. In our evaluation, an average of 23% of sentences are filtered from discussions. Figure 2 illustrates the effect of meta-sentence filtering on a discussion's set of sentence.

## 3.2 Generative Cluster Labeling

Most cluster labeling approaches extract keywords or key phrases as labels, which limits their fluency. These approaches may also require training data acquisition for supervised learning. We formulate cluster labeling as an unsupervised abstractive summarization task. We experiment with prompt-based large language models in zero-shot and few-shot settings. This enables generalization across multiple domains, the elimination of supervised learning, and fluent cluster labels with higher readability in comparison to keywords or phrases.

We develop several prompt templates specifically tailored for different types of LLMs. For encoder-decoder models, we carefully develop appropriate prompts based on PromptSource (Bach et al., 2022), a toolkit that provides a comprehensive collection of natural language prompts for various tasks across 180 datasets. In particular, we analyze prompts for text summarization datasets with respect to (1) descriptive words for the generation of cluster labels using abstractive summarization, (2) commonly used separators to distinguish instructions from context, (3) the position of instructions within prompts, and (4) the granularity level of input data (full text, document title, or sentence). Since our task is about summarizing groups of sentences, we chose prompts that require the full text as input to ensure that enough contextual

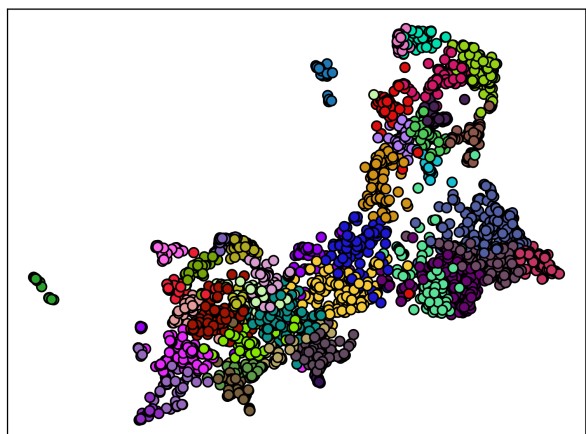

(a) Joint clustering of a discussion and meta-sentences $D \cup M'$.

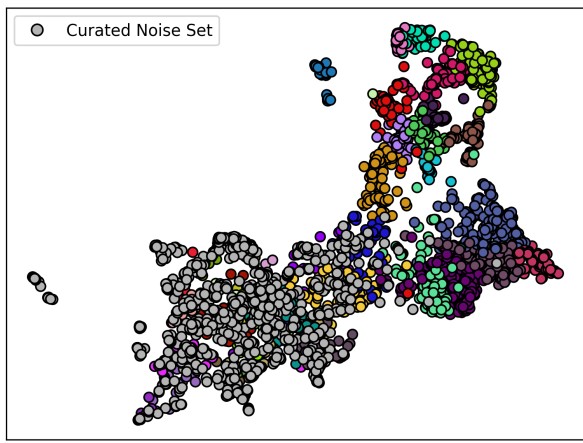

(b) The sampled meta-sentences $M' \subset M$ highlighted gray.

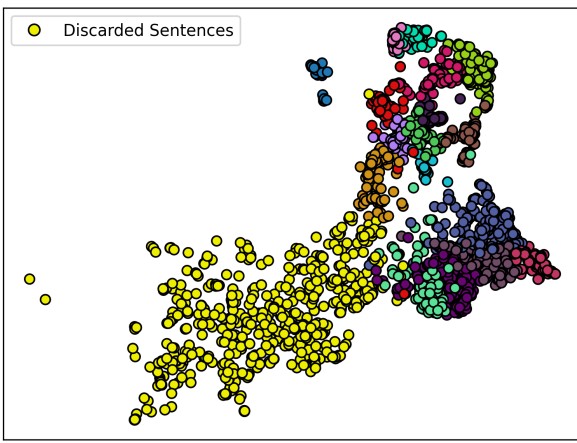

(c) Classification of meta-sentence clusters to be omitted.

Figure 2: Effect of meta-sentence filtering: (a and b) A discussion's sentences $D$ are jointly clustered with a sample of meta-sentences $M' \subset M$. (c) Then each cluster is classified as a meta-sentence cluster based on its proportion of meta-sentences and its neighboring clusters. Meta-sentence clusters are omitted.

information is provided (within the limits of each model's input size). Section 4.1 provides details on the prompt engineering process.

### 3.3 Frame Assignment

Any controversial topic can be discussed from different perspectives. For example, "the dangers of social media" can be discussed from a moral or a health perspective, among others. In our indicative summaries, we use argumentation frame labels as top-level headings to operationalize different perspectives. An argumentation frame may include one or more groups of relevant arguments. We assign frame labels from the issue-generic frame inventory shown in Table 1 (Boydstun et al., 2014) to each cluster label derived in the previous step.[7]

We use prompt-based models in both zero-shot and few-shot settings for frame assignment. In our experiments with instruction-tuned models, we designed two types of instructions, shown in Figure 10, namely direct instructions for models trained on instruction–response samples, and dialog instructions for chat models. The instructions are included along with the cluster labels in the prompts. Moreover, including the citation of the frame inventory used in our experiments has a positive effect on the effectiveness of some models (see Appendix D.1 for details).

### 3.4 Indicative Summary Presentation

Given the generated labels of all sentence clusters and the frame labels assigned to each cluster label, our indicative summary groups the cluster labels by their respective frame labels. The cluster label groups of each frame label are then ordered by cluster size. This results in a two-level indicative summary, as shown in Figures 1 and 4.

## 4 Prompt Engineering

Using prompt-based LLMs for generative cluster labeling and frame assignment requires model-specific prompt engineering as a preliminary step. We explored the 19 model variants listed in Table 2. To select the most appropriate models for our task, we consulted the HELM benchmark (Liang et al., 2022), which compares the effectiveness of different LLMs for different tasks. Further, we have included various recently released open source models (with optimized instructions) as they were released. Since many of them were released during our research, we reuse prompts previously optimized prompts for the newer models.[8]

---

[7]For detailed label descriptions see Table 6 in the Appendix.
[8]See Appendices C and D for details.

---

| Frame Inventory | |
|---|---|
| Capacity & Resources | Fairness & Equality |
| Constitutionality & Jurisprudence | Health & Safety |
| | Morality |
| Crime & Punishment | Policy Prescription & Evaluation |
| Cultural Identity | Political |
| Economic | Public Opinion |
| External Regulation & Reputation | Quality of Life |
| | Security & Defense |

Table 1: Inventory of frames proposed by Boydstun et al. (2014) to track the media's framing of policy issues.

| Model Variants | Description |
|---|---|
| **`Pre-InstructGPT`** | |
| **T0** vanilla | Encoder-decoder model trained on datasets transformed as task-specific prompts. |
| **BLOOM** vanilla | A multilingual autoregressive model with 176B parameters for prompt-based text completion. |
| **GPT-NeoX** 20B | Open source alternative to GPT-3. |
| **OPT** 66B | Autoregressive model with similar effectiveness to GPT-3, but more efficient data collection and training. |
| **`Direct Instruction`** | |
| **LLaMA-CoT** vanilla | LLaMA-30B fine-tuned on chain-of-thought and reasoning samples (Si and Lin, 2023). |
| **Alpaca** 7B | LLaMA-7B fine-tuned based on 52k self-instruct responses (Wang et al., 2022). |
| **OASST** vanilla | LLaMA-30B fine-tuned on the OpenAssistant Conversations dataset (Köpf et al., 2023) using reinforcement learning. |
| **Pythia** 12B | Suite of LLMs trained on public data to investigate the effects of training and scaling on various model properties. |
| **GPT\*** 3.5, Chat, 4 | OpenAI models GPT3.5 (*text-davinci-003*), ChatGPT (*gpt-3.5-turbo*), and GPT4. |
| **`Dialogue Instruction`** | |
| **LLaMA** 30B, 65B | Suite of open-source LLMs from Meta AI trained on public datasets. |
| **Vicuna** 7B, 13B | LLaMA models fine-tuned using conversations collected by ShareGPT (https://sharegpt.com) |
| **Baize** 7B, 13B | Open source chat model trained on 100k dialogues generated by letting ChatGPT (GPT 3.5-turbo) talk to itself. |
| **Falcon** 40B, 40B-Instruct | Trained on the RefinedWeb corpus (Penedo et al., 2023), which was obtained by filtering and deduplication of public web data. |

Table 2: LLMs studied for cluster labeling and frame assignment. Older models are listed by `Pre-InstructGPT` (prior to GPT3.5) and newer models are listed by their respective prompt types investigated (`Direct` / `Dialogue`). See Appendices C and D for details.

```
┌─────────────────────────────────────────┐
│ GPT3.5 for Generative Cluster Labeling   │
├─────────────────────────────────────────┤
│                                          │
│ Generate a single descriptive phrase     │
│ that describes the following debate in    │
│ very simple language, without talking     │
│ about the debate or the author.           │
│ Debate: """{text}"""                      │
│                                          │
└─────────────────────────────────────────┘

┌─────────────────────────────────────────┐
│ GPT4 for Frame Assignment                 │
├─────────────────────────────────────────┤
│                                          │
│ The following {input_type}ᵃ contains all  │
│ available media frames as defined in the  │
│ work from {authors}: {frames} For every   │
│ input, you answer with three of these     │
│ media frames corresponding to that input, │
│ in order of importance.                    │
│ ─────────────                             │
│ ᵃA list of frame labels or a JSON with    │
│   frame labels and their descriptions.    │
│                                          │
└─────────────────────────────────────────┘
```

Figure 3: The best performing instructions for cluster labeling and frame assignment. For frame assignment, citing the frame inventory using the placeholder {*authors*} has a positive impact on the effectiveness of some models (see Appendix D.1 for details).

### 4.1 Cluster Labeling

The prompts for the encoder-decoder model T0 are based on the PROMPTSOURCE (Bach et al., 2022) toolkit. We have experimented with different prompt templates and tried different combinations of input types (e.g. "text", "debate", "discussion", and "dialogue") and output types (e.g. "title", "topic", "summary", "theme", and "thesis"). The position of the instruction within a prompt was also varied, taking into account prefix and suffix positions. For decoder-only models like BLOOM, GPT-NeoX, OPT-66B, and OPT, we experimented with hand-crafted prompts. For GPT3.5, we followed the best practices described in OpenAI's API and created a single prompt.

Prompts were evaluated on a manually annotated set of 300 cluster labels using BERTScore (Zhang et al., 2020). We selected the most effective prompt for each of the above models for cluster labeling. Our evaluation in Section 5 shows that GPT3.5 performs best in this task. Figure 3 (top) shows the best prompt for this model.[9]

### 4.2 Frame Assignment

For frame assignment, models were prompted to predict a maximum of three frame labels for a given cluster label, ordered by relevance. Experiments were conducted with both direct instructions and dialogue prompts in zero-shot and few-shot set-

tings. In the zero-shot setting, we formulated three prompts containing (1) only frame labels, (2) frame labels with short descriptions, and (3) frame labels with full text descriptions (see Appendix D.2 for details). For the few-shot setting, we manually annotated up to two frames from the frame inventory of Table 1 for each of the 300 cluster labels generated by the best model GPT3.5 in the previous step. We included 42 examples (3 per frame) in the few-shot prompt containing the frame label, its full-text description, and three examples. The remaining 285 examples were used for subsequent frame assignment evaluation. Our evaluation in Section 5 shows that GPT4 performs best on this task. Figure 3 (bottom) shows its best prompt.

## 5 Evaluation

To evaluate our approach, we conducted automatic and manual evaluations focused on the cluster labeling quality and the frame assignment accuracy. We also evaluated the utility of our indicative summaries in a purpose-driven user study in which participants had the opportunity to explore long discussions and provide us with feedback.

### 5.1 Data and Preprocessing

We used the "Winning Arguments" corpus from Tan et al. (2016) as a data source for long discussions. It contains 25,043 discussions from the ChangeMyView Subreddit that took place between 2013 and 2016. The corpus was preprocessed by first removing noise replies and then meta-sentences. Noise replies are marked in the metadata of the corpus as "deleted" by their respective authors, posted by bots, or removed by moderators. In addition, replies that referred to the Reddit guidelines or forum-specific moderation were removed using pattern matching (see Appendix A for details). The remaining replies were split into a set of sentences using Spacy (Honnibal et al., 2020). To enable the unit clustering (of sentences) as described in Section 3.1, the set of meta-sentences $M$ is bootstrapped by first clustering the entire set of sentences from all discussions in the corpus and then manually examining the clusters to identify those that contain meta-sentences, resulting in $|M| = 955$ meta-sentences. After filtering out channel-specific noise, the (cleaned) sets of discussion sentences are clustered as described.

**Evaluation Data** From the preprocessed discussions, 300 sentence clusters were randomly se-

---

[9]ChatGPT and GPT4 were released after our evaluation.

| Model | Mean Rank | # First | Length | | |
|---|---|---|---|---|---|
| | | | Min | Max | Mean |
| GPT3.5 | **1.38** | **225** | 3 | 27 | 9.44 |
| BLOOM | 2.95 | 33 | 1 | 37 | 8.13 |
| GPT-NeoX | 3.20 | 20 | 1 | 34 | 7.42 |
| OPT | 3.36 | 12 | 1 | 30 | 8.27 |
| T0 | 3.72 | 28 | 1 | 18 | 3.10 |

Table 3: Results of the qualitative evaluation of generative cluster labeling. Shown are (1) the mean rank of a model from four annotators and (2) the number of times a model was ranked first by an annotator. GPT3.5 (*text-davinci-003*) performed better than other models and generated longer labels on average.

lected. Then, we manually created a cluster label and up to three frame labels for each cluster. Due to the short length of the cluster labels, up to two frames per label were sufficient. After excluding 57 examples with ambiguous frame assignments, we obtained a reference set of 243 cluster label samples, each labeled with up to two frames.

## 5.2 Generative Cluster Labeling

The results of the automatic cluster labeling evaluation using BERTScore and ROUGE are shown in (Appendix) Tables 7 and 8, respectively. We find that ChatGPT performs best. To manually evaluate the quality of the cluster labels, we used a ranking-based method in which four annotators scored the generated cluster labels against the manually annotated reference labels of each of the 300 clusters. To provide additional context for the cluster content, the five most semantically similar sentences to the reference label from each cluster were included, as well as five randomly selected sentences from the cluster. To avoid possible bias due to the length of the cluster labels by different models, longer labels were truncated to 15 tokens.[10] To determine an annotator's model ranking, we merged the preference rankings for all clusters using reciprocal rank fusion (Cormack et al., 2009). Annotator agreement was calculated using Kendall's $W$ for rank correlation (Kendall, 1948), which yielded a value of 0.66, indicating *substantial* agreement.

The average ranking of each model is shown in Table 3 along with the length distributions of the generated cluster labels.[11] GPT3.5 showed supe-

| Model | Zero-Shot | | | Few-Shot |
|---|---|---|---|---|
| | – | *short* | *full* | |
| `Alpaca-7B` | 39.1 | 39.5 | 28.4 | 20.6 |
| `Baize-7B` | 34.2 | 34.6 | 39.1 | 30.9 |
| `Baize-13B` | 42.4 | 48.1 | 42.0 | 39.5 |
| `BLOOM` | 26.7 | 31.7 | 25.5 | – |
| `ChatGPT` | 60.9[2] | 58.0[3] | 58.8[2] | 63.4[2] |
| `Falcon-40B` | 46.5 | 46.5 | 46.1 | 38.3 |
| `Falcon-40B-Inst.` | 51.4 | 44.4 | 32.9 | 28.4 |
| `GPT3.5` | 53.5[3] | **60.9**[1] | 58.0[3] | 53.9[4] |
| `GPT4` | **63.4**[1] | 60.5[2] | **65.4**[1] | **67.1**[1] |
| `GPT-NeoX` | 19.3 | 25.1 | 31.3 | 31.3 |
| `LLaMA-30B` | 45.7 | 41.2 | 39.1 | 40.7 |
| `LLaMA-CoT` | 46.9 | 54.3[4] | 49.8 | 57.2[3] |
| `LLaMA-65B` | 53.1[4] | 50.6[5] | 39.5 | – |
| `OASST` | 48.6[5] | 48.1 | 53.5[5] | 47.7 |
| `OPT` | 16.0 | 13.2 | 14.8 | – |
| `Pythia` | 31.7 | 33.3 | 30.5 | 29.6 |
| `T0` | 48.6[5] | 54.3[4] | 55.6[4] | 49.8[5] |
| `Vicuna-7B` | 28.4 | 36.2 | 35.4 | 20.2 |
| `Vicuna-13B` | 44.0 | 40.7 | 42.0 | 38.3 |

Table 4: Results of an automatic evaluation of 19 LLMs (sorted alphabetically) for frame assignment, indicating the five best models in each setting. Shown are the percentages of samples where the first frame predicted by a model is one of the reference frames. The three zero-shot columns denote the prompt type: frame label only, label with *short* description, and label with *full* description. Model types are also indicated: `Pre-InstructGPT`, `Direct` / `Dialogue`. Missing values are model inferences that exceed our computational resources.

rior effectivenss in generating high-quality cluster labels. It ranked first in 225 out of 300 comparisons, with an average score of 1.38 by the four annotators. The cluster labels generated by GPT3.5 were longer on average (9.4 tokens) and thus more informative than those generated by the other models, which often generated disjointed or incomplete labels. In particular, T0 generated very short labels on average (3.1 tokens) that were generic/non-descriptive.

## 5.3 Frame Assignment

In the zero-/few-shot frame assignment settings described in Section 4.2, we prompted the models to predict three frames per cluster label in order of relevance. Using the manually annotated reference set of 243 cluster labels and their frame labels, we evaluated the accuracy of the frames predicted for each cluster label that matched the reference frames. The results for the first predicted frame

---

[10]Figure 9 in the Appendix shows the annotation interfaces.

[11]As newer models were published after our manual evaluation, we show an automatic evaluation of all models using human and GPT3.5-based reference labels in the Appendix

in Tables 7 and 8.

are shown in Table 4. In most cases, GPT4 outperforms all other models in the various settings, with the exception of the zero-shot setting with a short prompt, where GPT3.5 narrowly outperforms GPT4 with 60.9% accuracy versus 60.5%. Among the top five models, the GPT* models that follow direct user instructions perform consistently well, with the LLaMA-/65B/-CoT and T0 models showing strong effectiveness among the open-source LLMs. Conversely, the OPT model performs consistently worse in all settings. The few-shot setting shows greater variance in results, suggesting that the models are more sensitive to the labeled examples provided in the prompts. Including a citation to the frame inventory paper in the instructions (see Figure 10) significantly improved the effectiveness of Falcon-40B (12%) and LLaMA-65B (9%) in the zero-shot setting (see Appendix D.1 for details).

### 5.4 Usefulness Evaluation

In addition to assessing each step of our approach, we conducted a user study to evaluate the effectiveness of the resulting indicative summaries. In this study, we considered two key tasks: *exploration* and *participation*. With respect to *exploration*, our goal was to evaluate the extent to which the summaries help users explore the discussion and discover new perspectives. With respect to *participation*, we wanted to assess how effectively the summaries enabled users to contribute new arguments by identifying the appropriate context and location for a response.

We asked five annotators to explore five randomly selected discussions from our dataset, for which we generated indicative summaries using our approach with GPT3.5. To facilitate intuitive exploration, we developed DISCUSSION EXPLORER (see Section 5.5), an interactive visual interface for the evaluated discussions and their indicative summaries. In addition to our summaries, two baselines were provided to annotators for comparison: (1) the original web page of the discussion on ChangeMyView, and (2) a search engine interface powered by Spacerini (Akiki et al., 2023). The search engine indexed the sentences within a discussion using the BM25 retrieval model. This allowed users to explore interesting perspectives by selecting self-selected keywords as queries, as opposed to the frame and cluster labels that our summaries provide. Annotators selected the best of these interfaces for exploration and participation.

**Results**   With respect to the *exploration* task, the five annotators agreed that our summaries outperformed the two baselines in terms of discovering arguments from different perspectives presented by participants. The inclusion of argumentation frames proved to be a valuable tool for the annotators, facilitating the rapid identification of different perspectives and the accompanying cluster labels showing the relevant subtopics in the discussion. For the *participation* task, three annotators preferred the original web page, while our summaries and the search engine were preferred by the remaining two annotators (one each) when it came to identifying the appropriate place in the discussion to put their arguments. In a post-study questionnaire, the annotators revealed that the original web page was preferred because of its better display of the various response threads, a feature not comparably reimplemented in DISCUSSION EXPLORER. The original web page felt "more familiar." However, we anticipate that this limitation can be addressed by seamlessly integrating our indicative summaries into a given discussion forum's web page, creating a consistent experience and a comprehensive and effective user interface for discussion participation.

### 5.5 DISCUSSION EXPLORER

Our approach places emphasis on summary presentation by structuring indicative summaries into a table of contents for discussions (see Section 3). To demonstrate the effectiveness of this presentation style in exploring long discussions, we have developed an interactive tool called DISCUSSION EXPLORER.[12] This tool illustrates how such summaries can be practically applied. Users can participate in discussions by selecting argumentation frames or cluster labels. Figure 4 presents indicative summaries generated by different models, providing a quick overview of the different perspectives. This two-level table of contents-like summary provides effortless navigation, allowing users to switch between viewing all arguments in a frame and understanding the context of sentences in a cluster of the discussion (see Figure 5).

## 6 Conclusion

We have developed an unsupervised approach to generating indicative summaries of long discussions to facilitate their effective exploration and navigation. Our summaries resemble tables of con-

---

[12]https://discussion-explorer.web.webis.de/

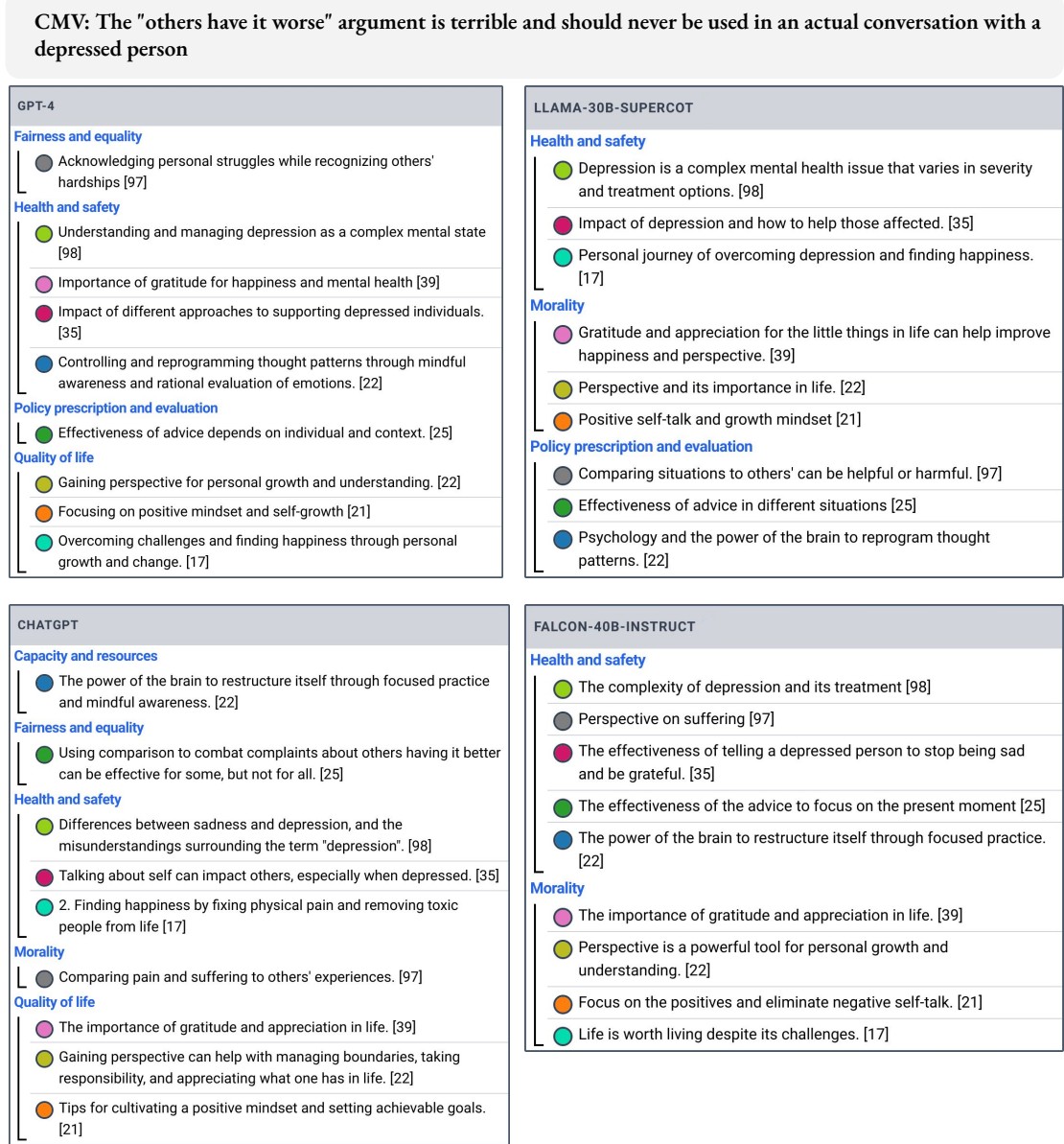

**CMV: The "others have it worse" argument is terrible and should never be used in an actual conversation with a depressed person**

**GPT-4**

**Fairness and equality**
- Acknowledging personal struggles while recognizing others' hardships [97]

**Health and safety**
- Understanding and managing depression as a complex mental state [98]
- Importance of gratitude for happiness and mental health [39]
- Impact of different approaches to supporting depressed individuals. [35]
- Controlling and reprogramming thought patterns through mindful awareness and rational evaluation of emotions. [22]

**Policy prescription and evaluation**
- Effectiveness of advice depends on individual and context. [25]

**Quality of life**
- Gaining perspective for personal growth and understanding. [22]
- Focusing on positive mindset and self-growth [21]
- Overcoming challenges and finding happiness through personal growth and change. [17]

**LLAMA-30B-SUPERCOT**

**Health and safety**
- Depression is a complex mental health issue that varies in severity and treatment options. [98]
- Impact of depression and how to help those affected. [35]
- Personal journey of overcoming depression and finding happiness. [17]

**Morality**
- Gratitude and appreciation for the little things in life can help improve happiness and perspective. [39]
- Perspective and its importance in life. [22]
- Positive self-talk and growth mindset [21]

**Policy prescription and evaluation**
- Comparing situations to others' can be helpful or harmful. [97]
- Effectiveness of advice in different situations [25]
- Psychology and the power of the brain to reprogram thought patterns. [22]

**CHATGPT**

**Capacity and resources**
- The power of the brain to restructure itself through focused practice and mindful awareness. [22]

**Fairness and equality**
- Using comparison to combat complaints about others having it better can be effective for some, but not for all. [25]

**Health and safety**
- Differences between sadness and depression, and the misunderstandings surrounding the term "depression". [98]
- Talking about self can impact others, especially when depressed. [35]
- 2. Finding happiness by fixing physical pain and removing toxic people from life [17]

**Morality**
- Comparing pain and suffering to others' experiences. [97]

**Quality of life**
- The importance of gratitude and appreciation in life. [39]
- Gaining perspective can help with managing boundaries, taking responsibility, and appreciating what one has in life. [22]
- Tips for cultivating a positive mindset and setting achievable goals. [21]

**FALCON-40B-INSTRUCT**

**Health and safety**
- The complexity of depression and its treatment [98]
- Perspective on suffering [97]
- The effectiveness of telling a depressed person to stop being sad and be grateful. [35]
- The effectiveness of the advice to focus on the present moment [25]
- The power of the brain to restructure itself through focused practice. [22]

**Morality**
- The importance of gratitude and appreciation in life. [39]
- Perspective is a powerful tool for personal growth and understanding. [22]
- Focus on the positives and eliminate negative self-talk. [21]
- Life is worth living despite its challenges. [17]

Figure 4: DISCUSSION EXPLORER provides a concise overview of indicative summaries from various models for a given discussion. The summary is organized hierarchically: The argument frames act as heading, while the associated cluster labels act as subheadings, similar to a table of contents. Cluster sizes are also indicated. Clicking on a frame lists all argument sentences in a discussion that assigned to that frame, while clicking on a cluster label shows the associated argument sentences that discuss a subtopic in the context of the discussion (see Figure 5).

tents, which list argumentation frames and concise abstractive summaries of the latent subtopics for a comprehensive overview of a discussion. By analyzing 19 prompt-based LLMs, we found that GPT3.5 and GPT4 perform impressively, with LLaMA fine-tuned using chain-of-thought being the second best. A user study of long discussions showed that our summaries were valuable for exploring and uncovering new perspectives in long discussions, an otherwise tedious task when relying solely on the original web pages. Finally, we presented DISCUSSION EXPLORER, an interactive visual tool designed to navigate through long discussions using the generated indicative summaries. This serves as a practical demonstration of how indicative summaries can be used effectively.

# 7  Limitations

We focused on developing a technology that facilitates the exploration of long, argumentative discussions on controversial topics. We strongly believe that our method can be easily generalized to other types of discussions, but budget constraints prevented us from exploring these as well. We also investigated state-of-the-art language models to summarize these discussions and found that commercial models (GPT3.5, GPT4) outperformed open-source models (LLaMA, T0) in generating indicative summaries. Since the commercial models are regularly updated, it is important to note that the results of our approach may differ in the future. Although one can define a fixed set of prompts for each model, our systematic search for the optimal prompts based on an evaluation metric is intended to improve the reproducibility of our approach as newer models are released regularly.

To evaluate the effectiveness of the generated summaries, we conducted a user study with five participants that demonstrated their usefulness in exploring discussions. Further research is needed on how to effectively integrate summaries of this type into discussion platform interfaces, which was beyond the scope of this paper.

## Acknowledgments

This work was partially supported by the European Commission under grant agreement GA 101070014 (OpenWebSearch.eu)

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

## A  Preprocessing

Deleted posts were matched using: "[deleted]", "[removed]", "[Wiki][Code][/r/DeltaBot]", "[History]". To remove posts from moderators, we used:

- "hello, users of cmv! this is a footnote from your moderators"

- "comment has been remove"

- "comment has been automatically removed"

- "if you would like to appeal, please message the moderators by clicking this link."

- "this comment has been overwritten by an open source script to protect"

- "then simply click on your username on reddit, go to the comments tab, scroll down as far as possibe (hint:use res), and hit the new overwrite button at the top."

- "reply to their comment with the delta symbol"

## B  Clustering Implementation

We employed HDBSCAN, a soft clustering algorithm (Campello et al., 2013) to cluster the contextual sentence embeddings from SBERT (Reimers and Gurevych, 2019). As these embeddings are high dimensional, we follow Grootendorst (2022) and apply dimensionality reduction on these embeddings via UMAP (McInnes et al., 2017) and cluster them based on their euclidean distance. Most parameters were selected according to official recommendations for UMAP,[13] and HDBSCAN.[14]

**UMAP Parameters**

**metric**   We set this to "cosine" because this is the natural metric for SBERT embeddings.

**n_neighbors**   We set this to 30 instead of the default value of 15 because this makes the reduction focus more on the global structure. This is important since the local structure is more sensitive to noise.

**n_components**   We set this value to 10.

**min_dist**   We set this value to 0 because this allows the points to be packed closer together which makes separating the clusters easier.

**HDBSCAN Parameters**

**metric**   We set this to "euclidean" because this the target metric that UMAP uses for reducing the points.

**cluster_selection_method**   We set this value to "leaf". An alternative choice for this options is "eom". This option has the tendency to create unreasonably large clusters. There are instances where it creates only two or three clusters even for very large discussions. The "leaf" method does not suffer from this problem but it is more dependent on the "min_cluster_size" parameter.

**min_cluster_size**   This parameter is the most important one for this approach. It is also not straight forward to find a value for this since the sizes of the main subtopics of a discussion depend on the size of the discussion. To find a good value, we sampled 50 discussion randomly and 50 discussion stratified by discussion length from all discussions. We compute the clustering for all 100

---

[13] https://umap-learn.readthedocs.io/en/latest/clustering.html
[14] https://hdbscan.readthedocs.io/en/latest/parameter_selection.html

Exploring the Discussion via an Indicative Summary

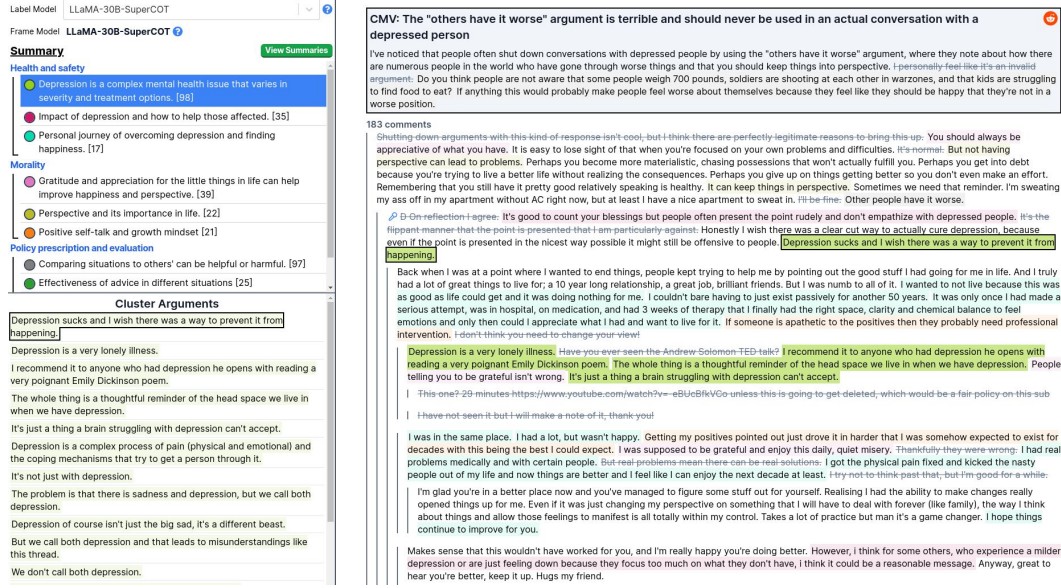

Figure 5: An exploratory view provided by DISCUSSION EXPLORER to quickly navigate a long discussion via an indicative summary. On the left, clicking on a cluster label lists all its constituent sentences. On the right, a specific sentence from the chosen cluster is presented in the context of the discussion. Softly highlighted are the sentences from other clusters that surround the selected sentence. Users can thus easily skim a discussion with several arguments for relevant information using the indicative summary in this exploratory view.

discussion for different values for min_cluster_size and manually determine a lower and upper bound for min_cluster_size that give a good clustering. We computed a regression model using the following function family as a basis: $f(x|a, b) = a \cdot x^b$ The input variable $x$ is the number of sentences in the discussion and the output variable is the average of the upper and lower bound. This yields the following function for computing min_cluster_size: $f(x) = 0.421 \cdot x^{0.559}$. Figure 6 visualizes upper and lower bounds as well as the found model.

## C  Generative Cluster Labeling

**Model Descriptions**   Given the large number of models investigated in the paper for both the tasks, we categorized them based on their release timelines. Models older than GPT3.5 are listed under `Pre-InstructGPT` such as T0, BLOOM, GPT-NeoX, and OPT. The `Direct` and `Dialogue` labels refer to models released after GPT3.5 which differ in their prompt styles as shown in Figure 7. Best prompts for the manually evaluated models (Section 5.2) are shown in Figure 8.

1. T0 (Sanh et al., 2022) is a prompt-based encoder-decoder model, fine-tuned on multiple tasks in-

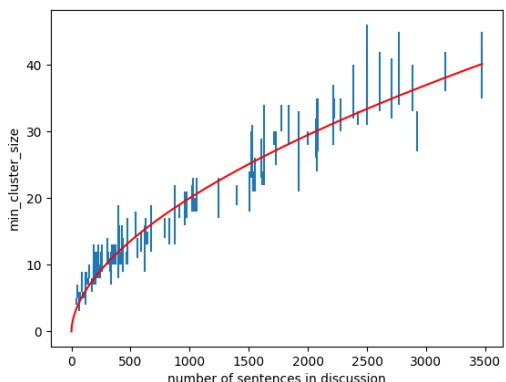

Figure 6: Blue vertical bars show the upper and lower bound for min_cluster_size that yield a good clustering for the corresponding discussion. The red curve shows the optimal fit for the regression.

cluding summarization, and surpasses GPT-3 in some tasks despite being much smaller. It was trained on prompted datasets where supervised datasets were transformed into prompts.

2. BLOOM (Scao et al., 2022) is an autoregressive LLM with 176B parameters, which specializes in prompt-based text completion for multiple languages. It also supports instruction-based

task completions for previously unseen tasks.

3. GPT-NeoX (Black et al., 2021) is an open-source, general-purpose alternative to the GPT-3 model (Ouyang et al., 2022) containing 20B parameters.

4. OPT (Zhang et al., 2022) is an autoregressive LLM with 66B parameters from the suite of decoder-only pre-trained transformers. These models offer similar performance and sizes as GPT-3 while employing more efficient practices for data collection and model training.

5. GPT3.5 (Brown et al., 2020; Ouyang et al., 2022) is an instruction-following LLM with 175B parameters that outperforms the GPT-3 model across several tasks by consistently adhering to user-provided instructions and generating high-quality, longer outputs. We used the *text-davinci-003* variant. In contrast to the other open-source models, it is accessible exclusively through the OpenAI API.[15]

**Prompt Descriptions**  We investigated several prompt templates for each model and selected the best performing one. All the prompts investigated for the encoder-decoder T0 model are shown in Table 11. Prompt templates for the decoder-only Pre-InstructGPT models (BLOOM, OPT, GPT-NeoX) are listed in Table 12. Prompt templates for the instruction-following LLMs are listed in Table 13.

**Automatic Evaluation**  For the sake of completion, we automatically evaluated the recently released (at the time of writing) instruction-following models. To adapt them to generative cluster labeling, we devised two instructions (Figure 7) similar to the direct and dialogue style instructions used for frame assignment (Section 3.3). Next, we computed BERTScore and ROUGE against two sets of references: (1) manually annotated ground-truth labels for 300 clusters, and (2) cluster labels from GPT3.5 which was the best model as per our manual evaluation (Section 5.2, Table 3). Complete results for BERTScore along with length distributions for the generated cluster labels are shown in Table 7, while results for ROUGE are shown in Table 8.

**Manual Evaluation**  Table 9 shows the guideline provided to the annotators. Figure 9 shows the

[15]https://platform.openai.com/docs/models/gpt-3-5

Figure 7: Direct and dialogue-style instructions for generative cluster labeling prompts. The best prompts for each model are shown in Figure 8.

annotation interface used to collect the rankings for cluster label quality.

## D  Frame Assignment

**Model Descriptions**  We categorize the models according to the instruction style followed for fine-tuning and generation. Instructions for each type are shown in Figure 10. The best prompts for each model are listed in Figure 11.

**Direct Instruction Models**

1. LLaMA-COT[16] is a finetuned model on datasets inducing chain-of-thought and logical deductions (Si and Lin, 2023).

2. Alpaca (Taori et al., 2023) is finetuned from the LLaMA 7B model (Touvron et al., 2023) using 52K self-instructed instruction-following examples (Wang et al., 2022).

3. OASST [17] is finetuned from LLaMA 30B on the OpenAssistant Conversations dataset (Köpf et al., 2023) using reinforcement learning.

4. Pythia (Biderman et al., 2023) is a suite of LLMs trained on public data to study the impact of training and scaling on various model properties. We used the 12B variant finetuned on the OpenAssistant Conversations dataset (Köpf et al., 2023).

5. GPT* includes models such as *text-davinci-003*, *gpt-3.5-turbo* (ChatGPT), and GPT-4 (Bubeck

[16]https://huggingface.co/ausboss/llama-30b-supercot
[17]https://huggingface.co/OpenAssistant/
oasst-rlhf-2-llama-30b-7k-steps-xor

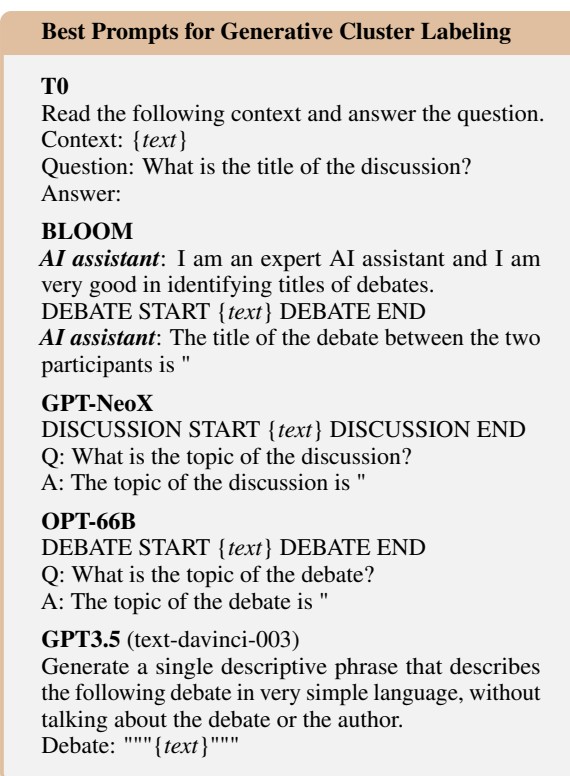

**Best Prompts for Generative Cluster Labeling**

**T0**
Read the following context and answer the question.
Context: {*text*}
Question: What is the title of the discussion?
Answer:

**BLOOM**
*AI assistant*: I am an expert AI assistant and I am very good in identifying titles of debates.
DEBATE START {*text*} DEBATE END
*AI assistant*: The title of the debate between the two participants is "

**GPT-NeoX**
DISCUSSION START {*text*} DISCUSSION END
Q: What is the topic of the discussion?
A: The topic of the discussion is "

**OPT-66B**
DEBATE START {*text*} DEBATE END
Q: What is the topic of the debate?
A: The topic of the debate is "

**GPT3.5** (text-davinci-003)
Generate a single descriptive phrase that describes the following debate in very simple language, without talking about the debate or the author.
Debate: """{*text*}"""

Figure 8: Best prompts for generative cluster labeling for each model. These prompts were chosen based on the automatic evaluation of several prompts for each model against 300 manually annotated cluster labels.

et al., 2023) from the OpenAI API. These models are not open-source but have demonstrated state-of-the-art performance across various tasks.

**Dialogue Instruction Models**

1. LLaMA (Touvron et al., 2023) is a suite of open-source LLMs trained on public datasets. We utilized the 30B and 65B variants.

2. Vicuna (Chiang et al., 2023) is finetuned from LLaMA using user-shared conversations collected from ShareGPT.[18] It has shown competitive performance when evaluated using GPT-4 as a judge. We used the 7B and 13B variants of this model.

3. Vicuna (Xu et al., 2023) is an open-source chat model trained on 100k dialogues generated by allowing ChatGPT (GPT 3.5-turbo) to converse with itself. We used the 7B and 13B variants of this model.

---

[18]https://sharegpt.com/

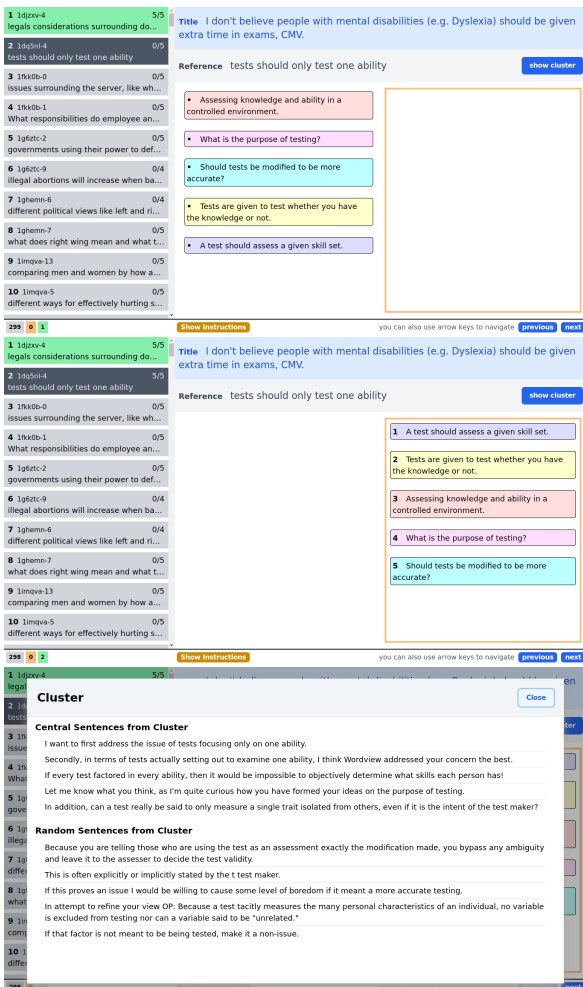

Figure 9: Annotation interface for ranking-based qualitative evaluation of cluster labels.

4. Falcon[19] is trained on the RefinedWeb dataset (Penedo et al., 2023), which is derived through extensive filtering and deduplication of publicly available web data. It is currently the state-of-the-art (at the time of writing) on the open-llm-leaderboard.[20] We utilized the 40B and 40B-Instruct variants of this model.

## D.1 Citation Impact on Frame Assignment

We conducted additional experiments to evaluate the impact of providing the citation of the media frames corpus paper by Boydstun et al. (2014) as additional information in the instructions shown in Section 3.3. This piece of information was provided after the substring "defined by" in the prompt template. Table 5 shows the results. We note that providing the citation information has a positive

---

[19]https://falconllm.tii.ae/
[20]https://huggingface.co/spaces/HuggingFaceH4/open_llm_leaderboard

_______________
[a] A list of frame labels or a JSON with frame labels and their descriptions.

Figure 10: Best performing instructions for frame assignment. Providing the citation for the frame inventory via the placeholder {*authors*} positively affects the performance of some models (Appendix D.1).

| Prompt | Falcon-40B | | ChatGPT | | LLaMA-65B | |
|---|---|---|---|---|---|---|
| | Cite. | – | Cite. | – | Cite. | – |
| **Zero-Shot** | 46.5 | 34.2 | 60.9 | 60.1 | 53.1 | 44.4 |
| **Zero-Shot (short)** | 46.5 | 42.8 | 58.0 | 57.2 | 50.6 | 42.4 |
| **Zero-Shot (full)** | 46.1 | 46.5 | 58.8 | 60.9 | 39.5 | 39.1 |
| **Few-Shot** | 38.3 | 39.1 | 63.4 | 64.6 | – | – |

Table 5: Analysis of the impact of providing citation of the media frames corpus paper as additional information in the instructions for the frame assignment. Providing citation information (**Cite.**) shows up to 12% improvement for **Falcon-40B** and 9% for **LLaMA-65B** under zero-shot setting (with only frame labels in the prompt).

impact on the performance of the models. The improvement is up to 12% for Falcon-40B and 9% for LLaMA-65B under zero-shot setting (only frame labels without descriptions in the prompt). This improvement can be attributed to the models being trained on a large text corpus, with the citation serving as a strong signal for generating more accurate labels. However, ChatGPT is only slightly affected.

## D.2 Zero-Shot and Few-Shot Prompts for Frame Assignment

### D.2.1 Zero-Shot (short)

```
[
  "economic",
  "capacity and resources",
  "morality",
  "fairness and equality",
  "legality, constitutionality and
↪   jurisprudence",
  "policy prescription and evaluation",
  "crime and punishment",
  "security and defense",
  "health and safety",
  "quality of life",
  "cultural identity",
  "public opinion",
  "political",
  "external regulation and reputation"
]
```

### D.2.2 Zero-Shot

```
{
  "economic": {
    "description": "costs, benefits, or other
↪   financial implications"
  },
  "capacity and resources": {
    "description": "availability of physical,
↪   human or financial resources, and
↪   capacity of current systems"
  },
  "morality": { "description": "religious or
↪   ethical implications" },
  "fairness and equality": {
    "description": "balance or distribution of
↪   rights, responsibilities, and resources"
  },
  "legality, constitutionality and
↪   jurisprudence": {
    "description": "rights, freedoms, and
↪   authority of individuals, corporations,
↪   and government"
  },
  "policy prescription and evaluation": {
    "description": "discussion of specific
↪   policies aimed at addressing problems"
  },
  "crime and punishment": {
    "description": "effectiveness and
↪   implications of laws and their
↪   enforcement"
  },
  "security and defense": {
    "description": "threats to welfare of the
↪   individual, community, or nation"
  },
  "health and safety": {
    "description": "health care, sanitation,
↪   public safety"
  },
  "quality of life": {
    "description": "threats and opportunities for
↪   the individual's wealth, happiness, and
↪   well-being"
  },
  "cultural identity": {
    "description": "traditions, customs, or
↪   values of a social group in relation to a
↪   policy issue"
  },
  "public opinion": {
    "description": "attitudes and opinions of the
↪   general public, including polling and
↪   demographics"
  },
  "political": {
```

```
    "description": "considerations related to
    ↪   politics and politicians, including
    ↪   lobbying, elections, and attempts to sway
    ↪   voters"
  },
  "external regulation and reputation": {
    "description": "international reputation or
    ↪   foreign policy of the U.S."
  }
}
```

### D.2.3 Zero-Shot (full)

```
{
  "economic": {
    "description": "The costs, benefits, or
    ↪   monetary/financial implications of the
    ↪   issue (to an individual, family,
    ↪   community, or to the economy as a
    ↪   whole)."
  },
  "capacity and resources": {
    "description": "The lack of or availability
    ↪   of physical, geographical, spatial,
    ↪   human, and financial resources, or the
    ↪   capacity of existing systems and
    ↪   resources to implement or carry out
    ↪   policy goals."
  },
  "morality": {
    "description": "Any perspective or policy
    ↪   objective or action (including proposed
    ↪   action) that is compelled by religious
    ↪   doctrine or interpretation, duty, honor,
    ↪   righteousness or any other sense of
    ↪   ethics or social responsibility."
  },
  "fairness and equality": {
    "description": "Equality or inequality with
    ↪   which laws, punishment, rewards, and
    ↪   resources are applied or distributed
    ↪   among individuals or groups. Also the
    ↪   balance between the rights or interests
    ↪   of one individual or group compared to
    ↪   another individual or group."
  },
  "legality, constitutionality and
  ↪   jurisprudence": {
    "description": "The constraints imposed on or
    ↪   freedoms granted to individuals,
    ↪   government, and corporations via the
    ↪   Constitution, Bill of Rights and other
    ↪   amendments, or judicial interpretation.
    ↪   This deals specifically with the
    ↪   authority of government to regulate, and
    ↪   the authority of individuals/corporations
    ↪   to act independently of government."
  },
  "policy prescription and evaluation": {
    "description": "Particular policies proposed
    ↪   for addressing an identified problem,
    ↪   and figuring out if certain policies will
    ↪   work, or if existing policies are
    ↪   effective."
  },
  "crime and punishment": {
```

```
    "description": "Specific policies in practice
    ↪   and their enforcement, incentives, and
    ↪   implications. Includes stories about
    ↪   enforcement and interpretation of laws by
    ↪   individuals and law enforcement,
    ↪   breaking laws, loopholes, fines,
    ↪   sentencing and punishment. Increases or
    ↪   reductions in crime."
  },
  "security and defense": {
    "description": "Security, threats to
    ↪   security, and protection of one's person,
    ↪   family, in-group, nation, etc. Generally
    ↪   an action or a call to action that can be
    ↪   taken to protect the welfare of a person,
    ↪   group, nation sometimes from a not yet
    ↪   manifested threat."
  },
  "health and safety": {
    "description": "Healthcare access and
    ↪   effectiveness, illness, disease,
    ↪   sanitation, obesity, mental health
    ↪   effects, prevention of or perpetuation
    ↪   of gun violence, infrastructure and
    ↪   building safety."
  },
  "quality of life": {
    "description": "The effects of a policy on
    ↪   individuals' wealth, mobility, access to
    ↪   resources, happiness, social structures,
    ↪   ease of day-to-day routines, quality of
    ↪   community life, etc."
  },
  "cultural identity": {
    "description": "The social norms, trends,
    ↪   values and customs constituting
    ↪   culture(s), as they relate to a specific
    ↪   policy issue."
  },
  "public opinion": {
    "description": "References to general social
    ↪   attitudes, polling and demographic
    ↪   information, as well as implied or actual
    ↪   consequences of diverging from or
    ↪   \"getting ahead of\" public opinion or
    ↪   polls."
  },
  "political": {
    "description": "Any political considerations
    ↪   surrounding an issue. Issue actions or
    ↪   efforts or stances that are political,
    ↪   such as partisan filibusters, lobbyist
    ↪   involvement, bipartisan efforts,
    ↪   deal-making and vote trading, appealing
    ↪   to one's base, mentions of political
    ↪   maneuvering. Explicit statements that a
    ↪   policy issue is good or bad for a
    ↪   particular political party."
  },
  "external regulation and reputation": {
    "description": "The United States' external
    ↪   relations with another nation; the
    ↪   external relations of one state with
    ↪   another; or relations between groups.
    ↪   This includes trade agreements and
    ↪   outcomes, comparisons of policy outcomes
    ↪   or desired policy outcomes."
  }
}
```

## D.2.4 Few-Shot

```
{
  "economic": {
    "description": "The costs, benefits, or
    ↪ monetary/financial implications of the
    ↪ issue (to an individual, family,
    ↪ community, or to the economy as a
    ↪ whole).",
    "examples": [
      "Necessity of minimum wage laws and their
      ↪ effects on the labor market.",
      "Consequences of unregulated capitalism and
      ↪ the potential of a libertarian
      ↪ society.",
      "Risk-based insurance premiums determined
      ↪ by complex modeling of probability and
      ↪ cost factors."
    ]
  },
  "capacity and resources": {
    "description": "The lack of or availability
    ↪ of physical, geographical, spatial,
    ↪ human, and financial resources, or the
    ↪ capacity of existing systems and
    ↪ resources to implement or carry out
    ↪ policy goals.",
    "examples": [
      "Potential of biofuels as an alternative to
      ↪ fossil fuels.",
      "Physical fitness tests measure upper body
      ↪ strength and running ability for
      ↪ military service.",
      "Physical strength and endurance needed for
      ↪ modern combat."
    ]
  },
  "morality": {
    "description": "Any perspective or policy
    ↪ objective or action (including proposed
    ↪ action) that is compelled by religious
    ↪ doctrine or interpretation, duty, honor,
    ↪ righteousness or any other sense of
    ↪ ethics or social responsibility.",
    "examples": [
      "Fighting for the weak and vulnerable
      ↪ despite the odds.",
      "Victim-blaming debate on police
      ↪ brutality.",
      "Potential corruption of some native
      ↪ canadian bands and the need for
      ↪ transparency."
    ]
  },
  "fairness and equality": {
    "description": "Equality or inequality with
    ↪ which laws, punishment, rewards, and
    ↪ resources are applied or distributed
    ↪ among individuals or groups. Also the
    ↪ balance between the rights or interests
    ↪ of one individual or group compared to
    ↪ another individual or group.",
    "examples": [
      "Differences between humanism and feminism
      ↪ and their respective goals.",
      "Disparities in scholarship opportunities
      ↪ for minority students.",
      "Violent suppression of native american
      ↪ populations for centuries leading to a
      ↪ lack of advocacy and rights."
    ]
  },
  "legality, constitutionality and
  ↪ jurisprudence": {
    "description": "The constraints imposed on or
    ↪ freedoms granted to individuals,
    ↪ government, and corporations via the
    ↪ Constitution, Bill of Rights and other
    ↪ amendments, or judicial interpretation.
    ↪ This deals specifically with the
    ↪ authority of government to regulate, and
    ↪ the authority of individuals/corporations
    ↪ to act independently of government.",
    "examples": [
      "Guns acquired through legal and illegal
      ↪ channels for criminal use.",
      "Importance of the 2nd amendment and the
      ↪ implications of gun ownership in a
      ↪ democracy.",
      "Relevance of sexual history in rape cases."
    ]
  },
  "policy prescription and evaluation": {
    "description": "Particular policies proposed
    ↪ for addressing an identified problem,
    ↪ and figuring out if certain policies will
    ↪ work, or if existing policies are
    ↪ effective.",
    "examples": [
      "Religious scientists making major
      ↪ contributions to the world despite
      ↪ majority of scientists being agnostic
      ↪ atheists.",
      "Pros and cons of voluntary registration.",
      "Collective ownership of production for the
      ↪ betterment of society, with workers
      ↪ profiting from the sale of their
      ↪ labor."
    ]
  },
  "crime and punishment": {
    "description": "Specific policies in practice
    ↪ and their enforcement, incentives, and
    ↪ implications. Includes stories about
    ↪ enforcement and interpretation of laws by
    ↪ individuals and law enforcement,
    ↪ breaking laws, loopholes, fines,
    ↪ sentencing and punishment. Increases or
    ↪ reductions in crime.",
    "examples": [
      "Complexities of police shootings and
      ↪ race.",
      "Men are more likely to commit violent
      ↪ crimes than women.",
      "Punishment as a response to crime debated,
      ↪ with consideration of morality,
      ↪ severity, and aims."
    ]
  },
  "security and defense": {
    "description": "Security, threats to
    ↪ security, and protection of one's person,
    ↪ family, in-group, nation, etc. Generally
    ↪ an action or a call to action that can be
    ↪ taken to protect the welfare of a person,
    ↪ group, nation sometimes from a not yet
    ↪ manifested threat.",
    "examples": [
      "Protective physical self-defense in a
      ↪ fight.",
```

```
        "Powerful military technology making
        ↪   infantry obsolete in war.",
        "Protection of infants and mentally
        ↪   disabled through social policy."
      ]
    },
    "health and safety": {
      "description": "Healthcare access and
      ↪   effectiveness, illness, disease,
      ↪   sanitation, obesity, mental health
      ↪   effects, prevention of or perpetuation
      ↪   of gun violence, infrastructure and
      ↪   building safety.",
      "examples": [
        "Complexities of food choices and their
        ↪   effects on health.",
        "Potentially fatal consequences of taking
        ↪   too much acetaminophen.",
        "Encouraging healthy habits without shaming
        ↪   or pressuring people to lose weight."
      ]
    },
    "quality of life": {
      "description": "The effects of a policy on
      ↪   individuals' wealth, mobility, access to
      ↪   resources, happiness, social structures,
      ↪   ease of day-to-day routines, quality of
      ↪   community life, etc.",
      "examples": [
        "Differences between adults and children in
        ↪   terms of understanding and
        ↪   perception.",
        "Importance of extracurriculars and
        ↪   academics for college admissions.",
        "Appropriate times to yell at customer
        ↪   service workers."
      ]
    },
    "cultural identity": {
      "description": "The social norms, trends,
      ↪   values and customs constituting
      ↪   culture(s), as they relate to a specific
      ↪   policy issue.",
      "examples": [
        "Rapid shift in acceptance of homosexuality
        ↪   in the u.s.",
        "Collective action necessary for social
        ↪   progress and change.",
        "Complexities of gender identity and
        ↪   expression."
      ]
    },
    "public opinion": {
      "description": "References to general social
      ↪   attitudes, polling and demographic
      ↪   information, as well as implied or actual
      ↪   consequences of diverging from or
      ↪   \"getting ahead of\" public opinion or
      ↪   polls.",
      "examples": [
        "Gender roles and expectations are socially
        ↪   constructed and changing.",
        "Pros and cons of the 40-hour work week.",
        "Potential appeal of a political
        ↪   candidate."
      ]
    },
    "political": {
      "description": "Any political considerations
      ↪   surrounding an issue. Issue actions or
      ↪   efforts or stances that are political,
      ↪   such as partisan filibusters, lobbyist
      ↪   involvement, bipartisan efforts,
      ↪   deal-making and vote trading, appealing
      ↪   to one's base, mentions of political
      ↪   maneuvering. Explicit statements that a
      ↪   policy issue is good or bad for a
      ↪   particular political party.",
      "examples": [
        "Differences between right-wing and
        ↪   left-wing politics.",
        "Complexities of anarchy.",
        "Power struggle between branches of
        ↪   government."
      ]
    },
    "external regulation and reputation": {
      "description": "The United States' external
      ↪   relations with another nation; the
      ↪   external relations of one state with
      ↪   another; or relations between groups.
      ↪   This includes trade agreements and
      ↪   outcomes, comparisons of policy outcomes
      ↪   or desired policy outcomes.",
      "examples": [
        "Implications of us involvement in nato and
        ↪   its allies.",
        "Potential consequences of us intervention
        ↪   in ukraine.",
        "Conflicting opinions on us involvement in
        ↪   foreign affairs."
      ]
    }
  }
}
```

## Best Prompts for Frame Assignment

**Alpaca-7B** (Direct Instruction)
Below is an instruction that describes a task, paired with an input that provides further context. Write a response that appropriately completes the request.
### Instruction:
{instruction}
### Input:
{input}
### Response:

**Vicuna-7B, 13B** (Dialogue Instruction)
{instruction}
USER: {input}
ASSISTANT:

**Pythia, OASST** (Direct Instruction)
<|system|>{instruction}<|endoftext|> <|prompter|>{input}<|endoftext|><|assistant|>

**LLaMA-30B, 65B** (Dialogue Instruction)
{instruction}
USER: {input}
ASSISTANT: ["

**LLaMA-CoT** (Direct Instruction)
Below is an instruction that describes a task, paired with an input that provides further context. Write a response that appropriately completes the request.
### Instruction:
{instruction}
### Input:
{input}
### Response:

**Falcon-40B, Instruct** (Dialogue Instruction)
{instruction}
USER: {input}
ASSISTANT: ["

**Baize-7B, 13B** (Dialogue Instruction)
{instruction}
$[|Human|]${input}
$[|AI|]$

**GPT3.5** (Direct Instruction)
{instruction}
Input: """ {input} """
Answer:

**ChatGPT** (Direct Instruction)

```
{
    "role": "system",
    "content": "{instruction}"
},
{
    "role": "user",
    "content": "{input}"
}
```

**GPT4** (Direct Instruction)

```
{
    "role": "system",
    "content": "{instruction}"
},
{
    "role": "user",
    "content": "{input}"
}
```

Figure 11: Best prompts for frame assignment for each model. The direct and dialogue instruction to be used with each prompt is shown in Figure 10.

| Frame | Description |
|-------|-------------|
| Capacity & Resources | The lack of or availability of physical, geographical, spatial, human, and financial resources, or the capacity of existing systems and resources to implement or carry out policy goals. |
| Constitutionality & Jurisprudence | The constraints imposed on or freedoms granted to individuals, government, and corporations via the Constitution, Bill of Rights and other amendments, or judicial interpretation. This deals specifically with the authority of government to regulate, and the authority of individuals/corporations to act independently of government. |
| Crime & Punishment | Specific policies in practice and their enforcement, incentives, and implications. Includes stories about enforcement and interpretation of laws by individuals and law enforcement, breaking laws, loopholes, fines, sentencing and punishment. Increases or reductions in crime. |
| Cultural Identity | The social norms, trends, values and customs constituting culture(s), as they relate to a specific policy issue. |
| Economic | The costs, benefits, or monetary/financial implications of the issue (to an individual, family, community or to the economy as a whole). |
| External Regulation & Reputation | A country's external relations with another nation; the external relations of one state with another; or relations between groups. This includes trade agreements and outcomes, comparisons of policy outcomes or desired policy outcomes. |
| Fairness & Equality | Equality or inequality with which laws, punishment, rewards, and resources are applied or distributed among individuals or groups. Also the balance between the rights or interests of one individual or group compared to another individual or group. |
| Health & Safety | Healthcare access and effectiveness, illness, disease, sanitation, obesity, mental health effects, prevention of or perpetuation of gun violence, infrastructure and building safety. |
| Morality | Any perspective—or policy objective or action (including proposed action)— that is compelled by religious doctrine or interpretation, duty, honor, righteousness or any other sense of ethics or social responsibility. |
| Policy Prescription & Evaluation | Particular policies proposed for addressing an identified problem, and figuring out if certain policies will work, or if existing policies are effective. |
| Political | Any political considerations surrounding an issue. Issue actions or efforts or stances that are political, such as partisan filibusters, lobbyist involvement, bipartisan efforts, deal-making and vote trading, appealing to one's base, mentions of political maneuvering. Explicit statements that a policy issue is good or bad for a particular political party. |
| Public Opinion | References to general social attitudes, polling and demographic information, as well as implied or actual consequences of diverging from or getting ahead of public opinion or polls. |
| Quality of Life | The effects of a policy, an individual's actions or decisions, on individuals' wealth, mobility, access to resources, happiness, social structures, ease of day-to-day routines, quality of community life, etc. |
| Security & Defense | Security, threats to security, and protection of one's person, family, in-group, nation, etc. Generally an action or a call to action that can be taken to protect the welfare of a person, group, nation sometimes from a not yet manifested threat. |
| Other | Any frames that do not fit into the above categories. |

Table 6: Descriptions of frames as per Boydstun et al. (2014). For zero-shot prompts, we experimented with providing (1) list of frames, (2) frames with relevant aspects from the descriptions (e.g. the *economic* frame has the aspects "costs", "benefits", "financial implications"), and (3) frames with complete descriptions as additional context.

| Model | Reference | | | GPT3.5 | | | Length | | |
|---|---|---|---|---|---|---|---|---|---|
| | **P** | **R** | **F1** | **P** | **R** | **F1** | **Min** | **Max** | **Mean** |
| Alpaca-7B | 0.20 | 0.15 | 0.17 | 0.31 | 0.28 | 0.29 | 3 | 21 | 7.92 |
| Baize-13B | 0.17 | 0.15 | 0.16 | 0.33 | 0.32 | 0.32 | 1 | 39 | 8.47 |
| Baize-7B | 0.22[3] | 0.19 | 0.20 | 0.38[3] | 0.38 | 0.38[3] | 2 | 46 | 10.73 |
| BLOOM | 0.15 | 0.09 | 0.11 | 0.22 | 0.19 | 0.20 | 1 | 54 | 8.13 |
| Falcon-40B | 0.12 | 0.09 | 0.10 | 0.17 | 0.17 | 0.17 | 1 | 57 | 9.57 |
| Falcon-40B-Inst. | 0.22[3] | 0.18 | 0.20 | 0.34 | 0.32 | 0.33 | 2 | 33 | 9.34 |
| ChatGPT | 0.23[2] | **0.24**[1] | **0.23**[1] | 0.39[2] | **0.43**[1] | **0.41**[1] | 3 | 34 | 11.10 |
| GPT4 | 0.21 | 0.19 | 0.20 | 0.37 | 0.36 | 0.37 | 4 | 18 | 7.50 |
| GPT-NeoX | 0.19 | 0.07 | 0.12 | 0.24 | 0.17 | 0.20 | 1 | 34 | 7.42 |
| LLaMA-30B | 0.12 | 0.06 | 0.08 | 0.19 | 0.17 | 0.17 | 1 | 46 | 9.58 |
| LLaMA-CoT | **0.24**[1] | 0.21[2] | 0.22[2] | **0.41**[1] | 0.39[3] | 0.40[2] | 3 | 29 | 8.45 |
| LLaMA-65B | 0.08 | 0.02 | 0.05 | 0.14 | 0.14 | 0.14 | 1 | 46 | 10.27 |
| OASST | 0.22[3] | 0.21[2] | 0.21[3] | 0.39[2] | 0.40[2] | 0.40[2] | 3 | 31 | 10.15 |
| OPT | 0.16 | 0.09 | 0.12 | 0.22 | 0.19 | 0.20 | 1 | 30 | 8.27 |
| Pythia | 0.19 | 0.13 | 0.16 | 0.31 | 0.27 | 0.29 | 2 | 34 | 7.69 |
| T0 | 0.15 | 0.00 | 0.06 | 0.15 | 0.03 | 0.09 | 1 | 18 | 3.10 |
| GPT3.5 | 0.23[2] | 0.20[3] | 0.21[3] | – | – | – | 3 | 27 | 9.44 |
| Vicuna-13B | 0.21 | 0.21[2] | 0.21[3] | 0.36 | 0.39[3] | 0.37 | 3 | 39 | 11.87 |
| Vicuna-7B | 0.20 | 0.19 | 0.19 | 0.34 | 0.37 | 0.35 | 2 | 42 | 11.47 |

Table 7: Complete results of automatic evaluation via BERTScore for the cluster labeling task of all 19 LLMs. We compared them against the manually annotated reference and **GPT3.5**, the best model from our manual evaluation. The top three models are indicated for each metric. Similar to the ROUGE evaluation, we see a strong performance by **ChatGPT** and **LLaMA-CoT**. Also shown are the statistics of the length of the generated cluster labels (in number of tokens).

| Model | Reference | | | GPT3.5 | | |
|---|---|---|---|---|---|---|
| | **R-1** | **R-2** | **R-LCS** | **R-1** | **R-2** | **R-LCS** |
| Alpaca-7B | 13.89 | 3.10 | 12.65 | 19.98 | 6.08 | 18.05 |
| Baize-13B | 14.44 | 2.28 | 13.02 | 24.59 | 8.23 | 22.53 |
| Baize-7B | 17.40 | 2.88 | 14.95 | 26.35 | 9.43 | 23.89 |
| BLOOM | 12.52 | 2.52 | 11.34 | 13.10 | 3.74 | 12.20 |
| Falcon-40B | 14.30 | 3.06 | 13.26 | 13.08 | 3.49 | 11.92 |
| Falcon-40B-Inst. | 17.59 | 3.97[3] | 15.48[3] | 21.72 | 7.66 | 19.57 |
| ChatGPT | **20.15**[1] | **4.88**[1] | **17.42**[1] | **29.52**[1] | 10.99[2] | 25.75[2] |
| GPT4 | 16.43 | 2.84 | 14.42 | 27.76[3] | 9.57[3] | 24.80[3] |
| GPT-NeoX | 12.93 | 2.37 | 11.72 | 11.67 | 2.41 | 10.72 |
| LLaMA-30B | 12.30 | 2.60 | 11.19 | 12.07 | 2.70 | 11.14 |
| LLaMA-CoT | 18.91[2] | 4.50[2] | 16.83[2] | 28.94[2] | **11.69**[1] | **26.38**[1] |
| LLaMA-65B | 10.25 | 1.93 | 9.40 | 10.81 | 2.49 | 9.95 |
| OASST | 18.28[3] | 3.58 | 16.15 | 27.13 | 9.09 | 23.88 |
| OPT | 11.67 | 2.68 | 10.87 | 10.56 | 2.17 | 9.55 |
| Pythia | 14.78 | 2.99 | 13.23 | 21.64 | 6.44 | 19.67 |
| T0 | 9.80 | 2.01 | 9.61 | 7.64 | 1.70 | 7.52 |
| GPT3.5 | 16.82 | 2.96 | 14.61 | – | – | – |
| Vicuna-13B | 16.90 | 3.02 | 14.81 | 25.32 | 8.66 | 22.66 |
| Vicuna-7B | 17.04 | 2.62 | 14.81 | 23.88 | 7.42 | 20.87 |

Table 8: Complete results of automatic evaluation via ROUGE for the cluster labeling task of all 19 LLMs. We compared them against the manually annotated reference and **GPT3.5**, the best model from our manual evaluation. The top three models are indicated for each metric. We see that **ChatGPT** and **LLaMA-CoT** perform strongly across the board.

**Guideline for judging the quality of the clustering**

**Task:** Given a reference text and a set of hypotheses, rank the hypotheses based on how similar they are to the reference text.

**How similar are the small texts to the reference text?**

Drag and drop the boxes with the texts on the left and bring them in your preferred order on the right. The most preferred text is on the top and the less you prefer a text, the lower it should be in the ranking.

Similarity is less in a sense of exact meaning but much rather in a meaning of is there some relation between the reference and hypotheses.

To get a better understanding of the meaning of the reference, the title of the original discussion and some central sentences from the cluster are provided (click the "show cluster" button next to the reference). The central sentences are selected based on how central they are in the original cluster and their mean similarity to the reference and hypotheses. So these are not perfectly representative to the cluster, but they can help you to get a better understanding of some hard to understand meanings.

Recommended Strategy for judging:
    The relation between the reference and hypotheses is understandable:
        → only read the reference and the hypotheses
    The reference is a bit weird:
        → read the title to get a better idea in what context the reference is used
    The hypotheses are hard to understand:
        → read the central sentences from the cluster for more context
    The relation between the reference and hypotheses are not clear:
        → read the central and random sentences from the cluster

**Note:** We are looking for a label that sufficiently describes the content of a cluster of sentences. It is important to understand that the reference is not the perfect label but rather strongly representative of the cluster.

When a lot of hypotheses talk about something that is not in the reference, it is sensible to include this information in the reference (implicitly) to make it "complete" while ranking the hypotheses.

**Example:**

*Reference*: responsibilities between employee and employer

Majority of the given hypotheses mention: "the service industry"

*Updated Reference*: responsibilities between employee and employer in the service industry

In the end we are looking for the central meaning of the cluster and it is very likely that at least one model got the central meaning right and the task is to guess what model got the central meaning best based on what the reference suggests the best central meaning is.

Table 9: Guideline for judging the quality of the clustering.

| Model | Zero-Shot | | | Zero-Shot (*short*) | | | Zero-Shot (*full*) | | | Few-Shot | | |
|---|---|---|---|---|---|---|---|---|---|---|---|---|
| | top 1 | top 2 | top 3 | top 1 | top 2 | top 3 | top 1 | top 2 | top 3 | top 1 | top 2 | top 3 |
| `Alpaca-7B` | 39.1 | 53.9 | 64.2 | 39.5 | 51.0 | 64.6 | 28.4 | 37.4 | 57.2 | 20.6 | 26.7 | 49.4 |
| `BLOOM` | 26.7 | 46.5 | 53.5 | 31.7 | 52.7 | 57.6 | 25.5 | 51.9 | 60.1 | – | – | – |
| `Baize-13B` | 42.4 | 53.5 | 58.4 | 48.1 | 59.3 | 63.4 | 42.0 | 53.5 | 60.5 | 39.5 | 46.5 | 49.4 |
| `Baize-7B` | 34.2 | 44.4 | 52.7 | 34.6 | 46.9 | 56.8 | 39.1 | 46.5 | 53.9 | 30.9 | 38.3 | 45.7 |
| `Falcon-40B` | 46.5 | 68.3 | 72.0 | 46.5 | 67.5 | 75.7 | 46.1 | 56.8 | 64.2 | 38.3 | 53.5 | 68.3 |
| `Falcon-40B-Inst.` | 51.4 | 64.6 | 72.8 | 44.4 | 56.4 | 68.3 | 32.9 | 44.9 | 57.6 | 28.4 | 49.4 | 63.8 |
| `ChatGPT` | 60.9 | 76.1 | 86.4 | 58.0 | 78.6 | 88.5 | 58.8 | 76.1 | 84.8 | 63.4 | 80.2 | 90.1 |
| `GPT-4` | 63.4 | 82.3 | 91.8 | 60.5 | 84.4 | 90.1 | 65.4 | 83.1 | 90.5 | 67.1 | 84.8 | 88.5 |
| `GPT-NeoX` | 19.3 | 28.4 | 50.6 | 25.1 | 31.3 | 51.9 | 31.3 | 36.6 | 50.2 | 31.3 | 39.5 | 49.0 |
| `LLaMA-30B` | 45.7 | 63.0 | 70.8 | 41.2 | 57.2 | 65.4 | 39.1 | 58.0 | 66.3 | 40.7 | 70.0 | 77.8 |
| `LLaMA-CoT` | 46.9 | 73.3 | 84.0 | 54.3 | 75.7 | 85.6 | 49.8 | 71.2 | 82.3 | 57.2 | 70.0 | 77.0 |
| `LLaMA-65B` | 53.1 | 65.4 | 81.9 | 50.6 | 70.8 | 82.3 | 39.5 | 64.6 | 78.6 | – | – | – |
| `OASST` | 48.6 | 72.8 | 82.3 | 48.1 | 66.3 | 76.5 | 53.5 | 73.7 | 82.7 | 47.7 | 65.0 | 79.8 |
| `OPT-66B` | 16.0 | 18.9 | 43.2 | 13.2 | 16.5 | 45.3 | 14.8 | 18.1 | 45.7 | – | – | – |
| `Pythia` | 31.7 | 44.0 | 52.3 | 33.3 | 43.6 | 49.4 | 30.5 | 39.1 | 44.9 | 29.6 | 34.2 | 38.7 |
| `T0++` | 48.6 | 58.4 | 64.2 | 54.3 | 60.1 | 65.4 | 55.6 | 59.7 | 63.8 | 49.8 | 52.3 | 53.5 |
| `GPT3.5` | 53.5 | 74.1 | 81.9 | 60.9 | 65.4 | 66.7 | 58.0 | 58.8 | 59.7 | 53.9 | 57.6 | 58.0 |
| `Vicuna-13B` | 44.0 | 52.7 | 62.1 | 40.7 | 55.1 | 67.1 | 42.0 | 53.1 | 64.6 | 38.3 | 50.2 | 60.1 |
| `Vicuna-7B` | 28.4 | 34.6 | 50.2 | 36.2 | 48.1 | 61.3 | 35.4 | 42.8 | 55.1 | 20.2 | 24.3 | 46.1 |

Table 10: Complete results of automatic evaluation for the frame assignment task. Shown are the **% of examples** where the first, second, and third predicted frames by a model are one of the reference frames. For the zero-shot setting, values are shown for each of the prompt type: only frame label, label with *short* description, and label with *full* description. Missing values are model inferences that exceeded our computational resources.

---

**Prompt Templates for T0**

**prefix**
```
What {output_type} would you choose for the {input_type} below?
{text}
```

**postfix**
```
{text}
What {output_type} would you choose for the {input_type} above?
```

**prefix-postfix**
```
What {output_type} would you choose for the {input_type} below?
{text}
What {output_type} would you choose for the {input_type} above?
```

**short**
```
{input_type}:
{text}
{output_type}:
```

**explicit**
```
{input_type} START
{text}
{input_type} END
{output_type} OF THE {input_type}:
```

**question answering**
```
Read the following context and answer the question.
Context:
{text}
Question: What is the {output_type} of the {input_type}?
Answer:
```

Table 11: Prompt templates investigated for T0 model for generative cluster labeling.

**Prompt Templates for BLOOM, GPT-NeoX, OPT, GPT3.5**

**dialogue**
```
AI assistant: I am an expert AI assistant. How can I help you?
Human: Can you tell me what the {output_type} of the following {input_type} is?
{input_type} START
{text}
{input_type} END
AI assistant: The {output_type} of the {input_type} is "
```

**explicit**
```
{input_type} START
{text}
{input_type} END
{output_type} of the {input_type}: "
```

**assistant solo**
```
AI assistant: I am an expert AI assistant and I am very good in identifying
↪ {output_type} of debates.
{input_type} START
{text}
{input_type} END
AI assistant: The {output_type} of the {input_type} between the two participants
↪ is "
```

**question answering**
```
{input_type} START
{text}
{input_type} END
Q: What is the {output_type} of the {input_type}?
A: The {output_type} of the {input_type} is "
```

**GPT3.5**
```
Generate a single descriptive phrase that describes the following debate in very
↪ simple language, without talking about the debate or the author.
Debate: """{text}"""
```

Table 12: Prompt templates investigated for generative cluster labeling with the four decoder-only models. The input_type is either "debate" or "discussion" and the output_type is either "title" or "topic".

| Prompt Templates for Instruction-following LLMs |
| --- |

**GPT3.5**
```
{instruction}
Input: """{input}"""
Answer:
```

**Alpaca-7B, LLaMA-CoT**
```
Below is an instruction that describes a task, paired with an input that provides
↪ further context. Write a response that appropriately completes the request.
### Instruction:
{instruction}
### Input:
{input}
### Response:
```

**Baize-13B, Baize-7B**
```
{instruction}
[|Human|]{input}
[|AI|]
```

**BLOOM, Falcon-40B, Falcon-40B-Instruct, GPT-NeoX, LLaMA-30B, LLaMA-65B, OPT-66B, Vicuna-13B, Vicuna-7B**
```
{instruction}
USER: {input}
ASSISTANT:
```

**OASST, Pythia**
```
<|system|>{instruction}<|endoftext|><|prompter|>{input}<|endoftext|><|assistant|>
```

**T0++**
```
{instruction}
Input: {input}
```

Table 13: Prompt templates used for experiments with instruction-following model.

| CMV: The "others have it worse" argument is terrible and should never be used in an actual conversation with a depressed person 
 Indicative Summary (LLaMA-CoT) |
| --- |

**Health & Safety**

- Depression is a complex mental health issue that varies in severity and treatment options. **[98]** (Policy Prescription & Evaluation)
- Impact of depression and how to help those affected. **[35]** (Morality)
- Personal journey of overcoming depression and finding happiness. **[17]** (Quality of Life)

**Morality**

- Gratitude and appreciation for the little things in life can help improve happiness and perspective. **[39]** (Quality of Life)
- Perspective and its importance in life. **[22]** (Fairness & Equality)
- Positive self-talk and growth mindset **[21]** (Fairness & Equality)

**Policy Prescription & Evaluation**

- Comparing situations to others' can be helpful or harmful. **[97]** (Morality)
- Effectiveness of advice in different situations **[25]** (Capacity & Resources)
- Psychology and the power of the brain to reprogram thought patterns. **[22]** (Morality)

Table 14: Indicative Summary from LLaMA-CoT.

**CMV: The "others have it worse" argument is terrible and should never be used in an actual conversation with a depressed person**
**Indicative Summary (GPT3.5)**

**Fairness & Equality**

- Complexities of comparing one's own struggles to those of others. **[97]** (Quality of Life)
- Advice can be helpful or unhelpful depending on how it is used. **[25]** (Morality)
- Focusing on personal goals and eliminating negative self-talk to create a growth mindset. **[21]** (Quality of Life)

**Health & Safety**

- How to help those with depression. **[35]** (Quality of Life)

**Morality**

- Differences between sadness and depression. **[98]** (Quality of Life)
- Reflecting on blessings and practicing gratitude to increase happiness. **[39]** (Quality of Life)
- Mindful awareness and reprogramming of thought patterns to take charge of emotions. **[22]** (Quality of Life)
- Gaining perspective to appreciate life and understand how one's actions affect others. **[22]** (Fairness & Equality)
- A journey of self-discovery and growth through difficult times. **[17]** (Quality of Life)

Table 15: Indicative Summary from GPT3.5.

**CMV: The "others have it worse" argument is terrible and should never be used in an actual conversation with a depressed person**
**Indicative Summary (GPT4)**

**Fairness & Equality**

- Acknowledging personal struggles while recognizing others' hardships **[97]** (Quality of Life)

**Health & Safety**

- Understanding and managing depression as a complex mental state **[98]** (Quality of Life)
- Importance of gratitude for happiness and mental health **[39]** (Quality of Life)
- Impact of different approaches to supporting depressed individuals. **[35]** (Quality of Life)
- Controlling and reprogramming thought patterns through mindful awareness and rational evaluation of emotions. **[22]** (Quality of Life)

**Policy Prescription & Evaluation**

- Effectiveness of advice depends on individual and context. **[25]** (Quality of Life)

**Quality of Life**

- Gaining perspective for personal growth and understanding. **[22]** (Morality)
- Focusing on positive mindset and self-growth **[21]** (Health & Safety)
- Overcoming challenges and finding happiness through personal growth and change. **[17]** (Morality)

Table 16: Indicative Summary from GPT4.

**CMV: Today is the best time period in human history to be alive for the vast majority of people.**
**Indicative Summary (LLaMA-CoT)**

**Capacity & Resources**
- The importance of having a private space for studying and building projects. **[33]** (Quality of Life)

**Crime & Punishment**
- Crime rates have changed over time. **[60]** (Security & Defense)

**Cultural Identity**
- Nostalgia for the 90s **[82]** (Quality of Life)

**Economic**
- Housing affordability is a complex issue with many factors at play. **[203]** (Capacity & Resources)
- Global poverty has decreased significantly over the past few decades. **[135]** (Capacity & Resources)
- Global trends and perspectives **[48]** (Policy Prescription & Evaluation)

**Health & Safety**
- AIDS pandemic was more fatal than the current one. **[97]** (Capacity & Resources)
- Current mental health epidemic and its causes. **[32]** (Capacity & Resources)

**Policy Prescription & Evaluation**
- Climate change is a serious issue that needs to be addressed. **[113]** (Economic)
- Concentration of military and economic power in history. **[47]** (Economic)

**Quality of Life**
- Best time period in human history to be alive. **[72]** (Economic)
- Quality of Life vs Expectations: Happiness Debate **[68]** (Other)
- Progress and improvement in society and culture **[51]** (Cultural Identity)
- Impact of technology on human connection and fulfillment. **[39]** (Cultural Identity)
- Middle Ages vs. Modern Times: Quality of Life Comparison **[29]** (Cultural Identity)

**Security & Defense**
- Statistics and data points in a debate about safety and progress **[43]** (Health & Safety)

Table 17: Indicative Summary from LLaMA-CoT.

**CMV: Today is the best time period in human history to be alive for the vast majority of people.**
**Indicative Summary (GPT3.5)**

### Crime & Punishment

- Violent crime rate has significantly decreased since the 1990s, but still remains an issue. **[60]** (Fairness & Equality)

### Cultural Identity

- A constant flow of information and societal changes causing a crisis of meaning. **[39]** (Quality of Life)

### Economic

- Catastrophic climate change leading to economic and ecological collapse. **[113]** (Health & Safety)
- Fragmented global economic and military power. **[47]** (Security & Defense)

### Fairness & Equality

- Housing prices have skyrocketed in the past decade, making it difficult for the average American to afford a home. **[203]** (Economic)
- Decrease in global poverty and hunger since the 90s, with a majority of the world population still living in poverty. **[135]** (Capacity & Resources)
- Differences between the 90s and the 2000s, and the effects of time periods on different generations. **[82]** (Quality of Life)
- Making progress towards a better world for future generations. **[51]** (Quality of Life)
- Throwing around statistics without meaning and misusing percentages. **[43]** (Policy Prescription & Evaluation)
- Room to study and compete in the job market. **[33]** (Capacity & Resources)
- A comparison of the lifestyles of lower-class people in the Middle Ages and modern times. **[29]** (Quality of Life)

### Health & Safety

- Effects of pandemics on population growth and life expectancy, with a comparison to the Bubonic Plague. **[97]** (Quality of Life)

### Morality

- Mental health crisis in the modern world and its potential causes. **[32]** (Quality of Life)

### Political

- Strong bias towards American perspective on global issues. **[48]** (Cultural Identity)

### Quality of Life

- Best time period in human history to be alive. **[72]** (Fairness & Equality)
- Balance between quality of life, expectations, and happiness, and how they relate to each other. **[68]** (Fairness & Equality)

Table 18: Indicative Summary from GPT3.5.

**CMV: Today is the best time period in human history to be alive for the vast majority of people.**
**Indicative Summary (GPT4)**

**Crime & Punishment**
- Violent crime rates have decreased since the 90s. **[60]** (Security & Defense)

**Cultural Identity**
- Nostalgia for the 90s and differing opinions on the era **[82]** (Quality of Life)
- Assuming most users are American **[48]** (Public Opinion)

**Economic**
- Housing affordability crisis in various locations **[203]** (Quality of Life)
- Reduced global poverty and hunger rates **[135]** (Fairness & Equality)
- Concentration of military and economic power in history **[47]** (Security & Defense)

**Health & Safety**
- Climate change and its worsening effects on Earth and humanity. **[113]** (Quality of Life)
- Comparing pandemics and death rates throughout history **[97]** (Quality of Life)
- Mental health awareness and treatment in modern society. **[32]** (Quality of Life)

**Policy Prescription & Evaluation**
- Acknowledging progress while recognizing room for improvement **[51]** (Quality of Life)

**Quality of Life**
- Best time to be alive debate **[72]** (Economic)
- Happiness influenced by expectations and quality of life. **[68]** (Economic)
- Misunderstanding and misuse of statistics **[43]** (Policy Prescription & Evaluation)
- Crisis of meaning and disconnection in modern society **[39]** (Cultural Identity)
- Importance of personal space for productivity and success **[33]** (Economic)
- Simple life in the Middle Ages vs modern lower class life **[29]** (Economic)

Table 19: Indicative Summary from GPT4.

**CMV: There shouldn't be anything other than the metric system.**
**Indicative Summary (LLaMA-CoT)**

**Capacity & Resources**

- Boiling and freezing points of water **[154]** (Quality of Life)

**Economic**

- Use of different size bottles in the dairy industry. **[87]** (Capacity & Resources)
- The cost of switching to the metric system is too high. **[59]** (Capacity & Resources)

**Health & Safety**

- Temperature ranges and weather conditions **[86]** (Quality of Life)
- Temperature range and clothing suggestions **[63]** (Quality of Life)

**Policy Prescription & Evaluation**

- Merits of Celsius and Fahrenheit temperature scales **[283]** (Constitutionality & Jurisprudence)
- Merits of the imperial and metric systems **[196]** (Constitutionality & Jurisprudence)
- Use of miles and feet in measuring distances **[140]** (Quality of Life)
- Merits of different systems of measurement **[106]** (Economic)
- Base 12 is better than base 10 for certain calculations. **[104]** (Capacity & Resources)
- Precision of measurements in inches and millimeters **[75]** (Quality of Life)
- Use of feet and inches for measuring height **[72]** (Constitutionality & Jurisprudence)
- Importance of precision in measurements **[64]** (Capacity & Resources)
- Metric vs. Imperial: Which system is better? **[52]** (Constitutionality & Jurisprudence)
- Merits of different counting systems **[49]** (Fairness & Equality)
- Merits of a decimal time system **[48]** (Economic)
- Merits of different scales and their practicality **[46]** (Capacity & Resources)
- Merits of different systems **[42]** (Economic)

Table 20: Indicative Summary from LLaMA-CoT.

**CMV: There shouldn't be anything other than the metric system.**
**Indicative Summary (GPT3.5)**

**Capacity & Resources**

- Over intuitive systems and their advantages. **[106]** (Quality of Life)
- For a more efficient counting system. **[104]** (Policy Prescription & Evaluation)
- Usefulness of different measurements for everyday use. **[87]** (Quality of Life)

**Economic**

- Legacy system rooted in society with benefits for everyday use and practical applications, but costly to transition away from. **[196]** (Fairness & Equality)
- Counting systems and their relative merits. **[49]** (Fairness & Equality)

**Fairness & Equality**

- Comparing the practicality of Celsius and Fahrenheit for everyday use, with no clear advantage to either. **[283]** (Quality of Life)
- Temperature scale based on water's freezing and boiling points. **[154]** (Constitutionality & Jurisprudence)
- Advantages and disadvantages of using inches and centimeters for measurements. **[75]** (Constitutionality & Jurisprudence)
- Usefulness of feet and inches for measuring human height. **[72]** (Quality of Life)
- A wide range of temperatures from chilly to hot, requiring different levels of clothing. **[63]** (Quality of Life)
- Costly transition to international standardization with little net benefit to average American. **[59]** (Economic)
- Advantages and disadvantages of the metric system. **[52]** (Constitutionality & Jurisprudence)
- Advantages and disadvantages of different scales. **[46]** (Capacity & Resources)
- Pros and cons of different systems. **[42]** (Policy Prescription & Evaluation)

**Health & Safety**

- Extremely cold temperatures ranging from -50C to +50C across the globe. **[86]** (Quality of Life)

**Constitutionality & Jurisprudence**

- Use of miles, yards, feet, and kilometers for measuring distances. **[140]** (Policy Prescription & Evaluation)
- Precision and accuracy in measurement. **[64]** (Policy Prescription & Evaluation)
- Complexities of measuring time. **[48]** (Policy Prescription & Evaluation)

Table 21: Indicative summary from GPT3.5.

**CMV: There shouldn't be anything other than the metric system.**
**Indicative Summary (GPT4)**

**Capacity & Resources**
- Water freezing and boiling points discussion **[154]** (Health & Safety)
- Base 12 system advantages **[104]** (Economic)
- Measurement units and their precision in various contexts **[75]** (Quality of Life)

**Cultural Identity**
- Preference for miles over kilometers in everyday language and distances **[140]** (Quality of Life)
- Preference based on familiarity and upbringing **[106]** (Quality of Life)
- Preference for feet and inches in measuring height **[72]** (Quality of Life)

**Economic**
- Costly and challenging transition to new system. **[59]** (Capacity & Resources)
- Using different counting systems and their efficiency in various situations. **[49]** (Capacity & Resources)

**Health & Safety**
- Temperature range discussion and its effects on daily life **[86]** (Quality of Life)
- Temperature and clothing preferences **[63]** (Quality of Life)

**Quality of Life**
- Comparing Celsius and Fahrenheit for everyday use **[283]** (Capacity & Resources)
- Imperial system vs. Metric system debate **[196]** (Cultural Identity)
- Metric and imperial measurements in daily life and their usefulness. **[87]** (Capacity & Resources)
- Misunderstanding precision and accuracy in measurements **[64]** (Health & Safety)
- Metric system advantages and precision debate **[52]** (Policy Prescription & Evaluation)
- Alternative time measurement systems **[48]** (Cultural Identity)
- Usefulness and subjectivity of different scales **[46]** (Fairness & Equality)
- Old system versus new system for everyday life **[42]** (Economic)

Table 22: Indicative summary from GPT4.

**CMV: Shoe sizes should be the same for both men and women**
**Indicative Summary (LLaMA-CoT)**

**Fairness & Equality**
- Men and women's feet are different in size and shape. **[73]** (Policy Prescription & Evaluation)
- Differences between men's and women's shoes and the impact of unisex shoes. **[44]** (Policy Prescription & Evaluation)
- Shoe sizes vary by sex due to differences in foot shape. **[30]** (Quality of Life)
- Women with broad but small feet struggle to find shoes that fit. **[19]** (Quality of Life)
- Differences in clothing proportions for men and women **[17]** (Policy Prescription & Evaluation)

**Policy Prescription & Evaluation**
- Pros and cons of standardizing shoe sizes **[64]** (Economic)
- Use of different measurement systems **[21]** (Economic)

Table 23: Indicative summary from LLaMA-CoT.

**CMV: Shoe sizes should be the same for both men and women**
**Indicative Summary (GPT3.5)**

**Fairness & Equality**
- Men and women's feet are differently shaped. **[73]** (Cultural Identity)
- Multiple shoe sizing systems causing confusion. **[64]** (Quality of Life)
- Gender-specific shoe design and comfort. **[44]** (Quality of Life)
- Different shoe sizes for men and women based on width and length. **[30]** (Quality of Life)
- Different measurement standards for length. **[21]** (Constitutionality & Jurisprudence)
- Wide feet struggle to find shoes that fit properly. **[19]** (Quality of Life)
- Clothing designed differently for men and women. **[17]** (Cultural Identity)

Table 24: Indicative summary from GPT3.5.

**CMV: Shoe sizes should be the same for both men and women**
**Indicative Summary (GPT4)**

**Economic**
- Shoe durability and gender differences in footwear preferences **[44]** (Quality of Life)

**Fairness & Equality**
- Standardizing shoe sizes for everyone **[64]** (Quality of Life)
- Differences in clothing proportions for men and women **[17]** (Cultural Identity)

**Health & Safety**
- Differences in men's and women's feet **[73]** (Quality of Life)

**Quality of Life**
- Shoe sizes differ for men and women due to width and shape differences in feet. **[30]** (Fairness & Equality)
- Different measurement systems for shoe sizes **[21]** (Cultural Identity)
- Finding shoes for wide and small feet **[19]** (Fairness & Equality)

Table 25: Indicative summary from GPT4.

**CMV: Social media is the most destructive addiction in our society**
**Indicative Summary (LLaMA-CoT)**

**Economic**
- Role of money in society and its impact on humanity. **[23]** (Capacity & Resources)
- Role of capitalism in society. **[20]** (Policy Prescription & Evaluation)
- Costs of running systems and offsetting those costs. **[19]** (Capacity & Resources)

**Fairness & Equality**
- Discrimination lawsuit against Amazon founder **[25]** (Constitutionality & Jurisprudence)

**Health & Safety**
- The impact of opioid addiction on individuals and society is devastating. **[107]** (Morality)

**Morality**
- Social media addiction vs opioid crisis **[38]** (Capacity & Resources)

**Policy Prescription & Evaluation**
- Pros and cons of social media and its impact on society. **[48]** (Morality)
- Measuring impact of technology on society **[35]** (Economic)
- The importance of education and community for a better world. **[26]** (Capacity & Resources)
- The impact of social media on society **[26]** (Public Opinion)
- The impact of social media on mental health is debated. **[24]** (Health & Safety)

Table 26: Indicative summary from LLaMA-CoT.

**CMV: Social media is the most destructive addiction in our society**
**Indicative Summary (GPT3.5)**

**Economic**
- Complexities of money as a social construct. **[23]** (Fairness & Equality)
- Costly infrastructure needed to run systems. **[19]** (Capacity & Resources)

**Fairness & Equality**
- Importance of education, societal injustices, and the consequences of comparing oneself to others. **[26]** (Quality of Life)
- Powerful man accused of denying bathroom access to employees. **[25]** (Constitutionality & Jurisprudence)
- Effects of capitalism on human behavior. **[20]** (Economic)

**Health & Safety**
- Effects of social media on mental health. **[24]** (Quality of Life)

**Morality**
- Devastating consequences of opioid addiction leading to death and destruction. **[107]** (Health & Safety)
- Effects of social media and opioid use on mental health. **[38]** (Health & Safety)
- Negative effects of social media outweigh the positives, leading to a lack of critical thinking and a moral panic. **[26]** (Quality of Life)

**Public Opinion**
- Pros and cons of social media. **[48]** (Cultural Identity)

**Quality of Life**
- Measuring societal impact through quality of life and direction of society. **[35]** (Cultural Identity)

Table 27: Indicative summary from GPT3.5.

---

**CMV: Social media is the most destructive addiction in our society**
**Indicative Summary (GPT4)**

**Economic**
- Money as a social construct and tool for exchange **[23]** (Quality of Life)
- Capitalism and human nature discussion **[20]** (Fairness & Equality)
- Costs and responsibilities of using resources and services **[19]** (Capacity & Resources)

**Health & Safety**
- Opioid crisis and its impact on individuals and society **[107]** (Quality of Life)
- Social media and opioid addiction relationship **[38]** (Quality of Life)
- Social media's impact on mental health and potential link to suicide rates. **[24]** (Quality of Life)

**Constitutionality & Jurisprudence**
- Lawsuit against Bezos for denying bathroom access **[25]** (Health & Safety)

**Quality of Life**
- Social media as a tool for connection and learning **[48]** (Capacity & Resources)
- Measuring impact through quality of life and societal direction **[35]** (Fairness & Equality)
- Improving society through better education and empathy. **[26]** (Fairness & Equality)
- Impact of social media on society and individuals **[26]** (Cultural Identity)

Table 28: Indicative summary from GPT4.