# OpenReview forum: "Indicative Summarization of Long Discussions"
_EMNLP/2023/Conference — EMNLP 2023 Main_

### Official Review · Reviewer_oFS8 · 2023-08-04

**Soundness:** 4

**Excitement:**

3: Ambivalent: It has merits (e.g., it reports state-of-the-art results, the idea is nice), but there are key weaknesses (e.g., it describes incremental work), and it can significantly benefit from another round of revision. However, I won't object to accepting it if my co-reviewers champion it.

**Paper Topic And Main Contributions:**

This paper presents a prompt-based zero-shot method to generate indicative summaries for long discussions. The authors introduce a three-step approach of clustering argument sentences by subtopics, generating a short summary that captures the gist of each cluster known as the cluster label, and then assigning a frame to each cluster label. Their results show that GPT3.5 outperforms all the other open-source models.

**Reasons To Accept:**

- Experimenting with 19 LLMs is remarkable, this allows comparison over a wide range of state-of-the-art models.
- Applying corpus-specific filtering and discussion-specific filtering is a reasonable approach, as this helps eradicate the less relevant sentences.
- The annotation interface for ranking the cluster-based quality seems useful and user-friendly.

**Reasons To Reject:**

- A few missing details in the paper: there should be a brief explanation as to how the annotators were selected. In section 5.2, four annotators are used for cluster labeling, however in section 5.4, we see five annotators. Are they the same annotators, what is the explanation for adding an extra annotator for this task?
- There seems to be a lot going on in Figure 1, it would be helpful to perhaps include what the number in parenthesis means or take it out.
- There are numerous metrics for automatic evaluation, why was only BERTScore used?

**Reproducibility:**

3: Could reproduce the results with some difficulty. The settings of parameters are underspecified or subjectively determined; the training/evaluation data are not widely available.

**Reviewer Confidence:**

4: Quite sure. I tried to check the important points carefully. It's unlikely, though conceivable, that I missed something that should affect my ratings.

**Typos Grammar Style And Presentation Improvements:**

- There are a few small typos in the paper. In section 3.2, there is a typo in the third sentence – it should say “been” and not “be”. Also, there is a repetition in the caption for Table 4.

---

> ### Author Rebuttal · Authors · 2023-08-26
>
> Thanks for your valuable feedback and pointing to the typos. We try to address your main concerns below.
>
> __Annotator Selection Choices.__
> Annotator selection for evaluating the generated cluster labels (4 annotators) and the final user study (5 annotators) was intentionally kept separate to prevent bias stemming from exposure to high-quality labels in the initial evaluation. To ensure neutrality, we engaged 5 new annotators for the user study, presenting the broader task of exploring lengthy discussions through a dedicated interface. We will provide further clarification on this distinction in the paper.
>
> __Figure Clarifications.__
> The number in parentheses is the number of sentences belonging to a given cluster(label). This was supposed to be clarified by the gray parenthesis “(replies; click label to view)” from a user’s perspective. In hindsight, this was too subtle. We will clarify this in the figure caption.
>
> __Usage of BERTScore.__
> We chose BERTScore due to its consideration of semantic similarity between generated cluster labels and human labels through contextual embeddings, a feature not captured by lexical metrics like ROUGE or BLEU.

---

### Official Review · Reviewer_UYXG · 2023-08-05

**Soundness:** 3

**Excitement:**

3: Ambivalent: It has merits (e.g., it reports state-of-the-art results, the idea is nice), but there are key weaknesses (e.g., it describes incremental work), and it can significantly benefit from another round of revision. However, I won't object to accepting it if my co-reviewers champion it.

**Missing References:**

- [Ref 1] Byeongchang Kim, Hyunwoo Kim, Gunhee Kim, Abstractive Summarization of Reddit Posts with Multi-level Memory Networks, NAACL 2019. https://aclanthology.org/N19-1260/

**Paper Topic And Main Contributions:**

This paper tackles the task of summarizing online forum discussion by making an abstractive summary from multiple sentences in the same group and assigning media frame(s) to each group, which the paper refers to as indicative summarization. The paper develops a multi-stage pipeline that first clusters multiple sentences, which are converted into concise summaries (called labels in the paper), and media frame(s) are assigned to each of the labels. Finally, the assigned frames are used to display summaries to the user. The paper evaluated the quality of cluster labeling and frame assignment.

**Questions For The Authors:**

Please respond to (W1)-(W4) above.

**Reasons To Accept:**

(S1) The paper tackles a multi-document summarization task from a slightly uncommon angle by formulating as indicative summarization.
(S2) The paper conducts comprehensive experiments using a variety of LLMs.
(S3) The paper is generally well-written and easy to follow.

**Reasons To Reject:**

(W1) (Task definition and novelty are not clear.) The notion of indicative summarization is not clear. According to Footnote 2, it is not clear what is different from extreme summarization (e.g., single-document such as XSum and multi-document such as TLDR). The TIFU Reddit dataset [Ref 1] is not cited or mentioned in this paper.

(W2) (Evaluation results do not verify the end-to-end task performance) The overall performance evaluation is missing or has some issues. Now the task consists of multiple components: (1) sentence clustering, (2) label assignment (which can be considered summarization), (3) frame assignment.

In 5.4, the author(s) compared with search-based solutions but summarization methods should also be compared as baselines. It is obvious that any summarization methods are better than search-based methods for exploration. And the results in 5.4 do not really support that the proposed solution is a better indicative summarization method than other summarization methods.

(W3) (Design choice is not well-justified; Technical novelty and significance are not clear) Similar to (W1) and (W2), it is not clear sentence clustering before frame assignments. Can frame assignment be done directly to each sentence before clustering? The lack of design choice discussion and ablation study makes it difficult to judge the justification of the design choice for the problem.  The paper used existing LLMs and existing media frames to tackle the problem. Technical novelty and significance are not clear from the paper in its current form.

(W4) (Comparisons of a variety of zero-shot LLMs with no clear implications) Although the paper conducts comprehensive experiments by using a variety of LLMs, the learnings are not clear. Specifically, Table 4 does not tell which LLMs we should use for the purpose (in this case, frame assignment). The paper does not offer training or validation datasets. The reported results can be this task & dataset-only and may not be generalized. Again, it’s difficult to judge only from the results.


### Minor comments

- LL269-272, the author(s) claim that it is novel to apply abstractive summarization models to clutter labeling. However, the problem itself is actually multi-sentence summarization, which the author(s) referred to as cluster labeling and I do not agree with this claim. (Also, extractive-abstractive can be considered clustering labeling models then.) I leave this comment as a minor comment as it is not directly related to the main claim(s).
- To me, the task looks closer to Argument Mining rather than Summarization. In any case, the paper should further clarify the differences against Argument Mining/Discussion Summarization.

**Reproducibility:**

3: Could reproduce the results with some difficulty. The settings of parameters are underspecified or subjectively determined; the training/evaluation data are not widely available.

**Reviewer Confidence:**

4: Quite sure. I tried to check the important points carefully. It's unlikely, though conceivable, that I missed something that should affect my ratings.

---

> ### Author Rebuttal · Authors · 2023-08-26
>
> Thank you for your detailed comments and questions. We answered them to the best of our ability below.
>
> ### W1: Task definition and novelty are not clear
>
> Thanks for pointing this out.  We acknowledge that the clarity of our task definition and novelty could have been improved. We provide a more detailed explanation below and will integrate it into the paper:
>
> __Task definition__
>
> Indicative summaries serve the purpose of "suggesting" the content within a discussion, aiding readers in determining whether they should delve deeper into the details. In contrast, informative summaries aim to (partially) "replace" the discussion by encapsulating the key information. We draw inspiration from Klaas (2005) “Toward Indicative Discussion Fora Summarization" (referenced in our paper) to establish a similar conceptual framework.
>
> __Novelty__
>
> Our approach introduces a new and completely unsupervised approach to generating indicative summaries that align with the discussion's structure, akin to a Table of Contents. These summaries guide readers through discussions instead of replacing them. We found no prior instances of generating such summaries for online discussions in existing literature. Moreover, our systematic prompt optimization based on established subtasks (summarization, paraphrasing, title generation), an extensive quantitative and qualitative evaluation of 19 LLMs, and a summary purpose-oriented user study are key contributions.
>
> ### W2: Evaluation results do not verify the end-to-end task performance
>
> We evaluated end-to-end performance, focusing on utility of indicative summaries. Our manual user study (Section 5.4) compared intuitive baselines.
>
> We chose search-based baselines to reflect common online reader behavior—employing (in-browser) keyword searches for finding relevant discussion content. A search engine allows users to find pertinent sentences and arguments using their own queries.
> Additionally, we provided the original web page of the discussion (from Reddit) for its presentation of reply threads to ensure study subjects have a familiar user experience.
>
> We avoided extractive summaries due to potential restrictions such as low coverage of interesting arguments, redundancy of similar arguments, and ambiguity in determining the optimal summary length for a given discussion. However, he search-based baselines mimic extractive summaries at the sentence level for the top-k most relevant results, providing for flexible summary lengths and avoiding these concerns.
>
> “__And the results in 5.4 do not really support that the proposed solution is a better indicative summarization method than other summarization methods.__”
>
> Annotators unanimously preferred our approach (lines 538 - 542) over baselines for uncovering new perspectives in conversations. Unlike other methods that generate _informative_ summaries to partially replace the discussions, our approach provides exploratory indicative summaries.
>
> ### W3: Design choice is not well-justified; Technical novelty and significance are not clear
>
> The decision to assign frames after clustering stems from careful consideration of practical issues:
>
> ___Handling Noise___: Online discussions often contain noisy sentences like meta comments, leading to irrelevant frame assignments when content lacks clear information. This could deteriorate summary quality due to subpar cluster labeling during abstractive summarization, especially if meta comments and noise dominate a cluster. Frame assignment after clustering and filtering such noise mitigates this issue.
>
> ___Efficient Cluster Labeling___: Focusing on assigning a frame for the cluster label itself, rather than hundreds of sentences, reduces model inferences. This streamlining enhances our approach's practical feasibility.
>
> Our indicative summaries mimic the table of contents in books, where "chapters" and "sections" outline hierarchy: frame labels serve as "chapters" and cluster labels as "sections".
>
> __Technical Novelty and Significance__
> We substantiate the novelty and importance of our contributions as follows:
>
> 1. ___Distinctive Summary Style___. The task of indicative summarization in general has received little attention compared to informative summarization of news or scholarly documents (as outlined in our related work). For a user wanting to follow up or participate in a discussion, the generated table of contents help them to quickly navigate the various arguments and learn about different perspectives. Moreover, such summaries have been mainly single texts (informative summaries) aimed at replacing the entire discussion rather than letting users explore them to discover new perspectives.
>
> 2. ___New Unsupervised Approach and Extensive Proof-of-Concept Evaluation for Indicative Summarization___.  We present a new approach to indicative summarization that is unsupervised and thus does not require large-scale labeled datasets. To show that both, the notion of indicative summaries and our approach are feasible, we conduct an extensive proof-of-concept evaluation in an important application domain. We deliberately choose to ground our approach in existing approaches rather than in addition also developing new ones, as this shows feasibility of the task of indicative summarization, and that this is achievable using the state of the art. Moreover, there was little evidence that state-of-the-art models were inappropriate; rather, our evaluation shows for the first time where indicative summarization can be improved in future work.  Finally, our evaluation of 19 LLMs to find the best models for this task, established the first and a reliable baseline for future work.
>
> 3. ___Instrumental Methods: Systematic Prompt Engineering, Noise Filtering, and Purpose-driven Evaluation___. Part of our approach includes original algorithmic contributions instead of reusing state-of-the-art approaches: We introduced a systematic prompt identification method (Section 4) for generative cluster labeling and frame assignment across different model classes (dialog vs. instruction-based), streamlining resource-intensive LLM investigations for the research community. Our clustering-based noise filtering offers an adaptable solution for diverse corpora. This avoids manually crafting several noise filtering patterns by manually looking at a large number of examples. Finally, our evaluation is contextually grounded, involving real-world tasks, unlike many other summarization assessments relying solely on automatic metrics.
>
> ### W4: Comparisons of LLMs with no clear implications
>
> We beg to differ. Clear conclusions are drawn in lines 455 - 460 for cluster labeling, lines 485 to 493 for frame assignment, and lines 570 to 573 in conclusion, about which models perform best for the task of indicative summarization. As stated in the paper, closed-source GPT models (ChatGPT, GPT4) perform best, followed by the LLaMa model combined with Chain-of-Thought prompting performing best among the open source LLMs. This implies that the research community can spare costly computational resources to investigate the other 16 LLMs for this task.
>
> The best models in Table 4 are indicated in bold for zero-shot and few-shot frame assignment. We will mention this in the table caption.
>
> “__The paper does not offer training or validation datasets__.”
> Regarding training and validation sets, our approach is completely unsupervised in that it does not need any training or validation sets. The discussion titles used for qualitative evaluation of usefulness and their corresponding summaries from the three best models are already provided in the appendix. The complete data used for evaluation and preprocessing will be made publicly available.
>
>  “__The reported results can be this task & dataset-only and may not be generalized__.”
> In this paper, we first thoroughly tackled the task on one specific and a highly relevant genre (Reddit discussions) and demonstrated via proof-of-concept that the approach is feasible and effective. To conclude generalizability across several genres/datasets would effectively require a significant amount of systematic evaluation (i.e., user studies, manual quality assessments, and error analyses tailored to each domain) that would not fit into the length restrictions of a typical research paper. However, follow up works targeted at  different domains can easily adapt the approach and evaluation design established in this paper.
>
> “__Again, it’s difficult to judge only from the results__.”
> We do not completely understand what is implied by this.
> With regards to our evaluation, we did not rely solely on automatic metrics to determine the best models (as is usually the case in summarization), but rather carefully designed, fine-grained, and resource-intensive annotation experiments for robust qualitative evaluation of the resulting summaries. In this regard, concrete suggestions as to what is still missing to draw conclusions would be useful.
>
> __Regarding low reproducibility score__:
> We have already included the complete code in the supplementary material, along with the annotation interfaces and the web application utilized for judgment collection. The appendix thoroughly explains all parameter values. The data will be made publicly available post-review, alongside the code. All of this has been checked for reproducibility and is easy to re-execute. If any crucial component has been inadvertently omitted, please inform us, and we will make every effort to address it.

---

### Official Review · Reviewer_RYVk · 2023-08-05

**Soundness:** 4

**Excitement:**

4: Strong: This paper deepens the understanding of some phenomenon or lowers the barriers to an existing research direction.

**Paper Topic And Main Contributions:**

The paper handles long discussion summarization by clustering sentences in the discussion and applying cluster labels using LLMs provide topic wise summary to the user.  They did a evaluation on 300 clusters in their system with a user study with 5 participants

**Questions For The Authors:**

1. What happens if the discussion is single topic - where all users discuss the same topic ? Did authors find such discussions how did your approach tackle those
2. How much does the corpus based filtering important - an ablation study would have provided more details about this filtering step as this makes the approach corpus dependent
3. While user study evaluates whole discussion the offline evaluation is only based on clusters ? Is it possible to rather select N discussions and clusters in them to understand how the performance works - this will provide a better understanding like the quality of clusters formed, with-in cluster accuracy and across cluster accuracy of a method

**Reasons To Accept:**

The paper is well written with comprehensive literature
The paper explains the preprocessing steps in details with example sentences for reader to understand well
The clustering, frame assignment and labels are explained in proper details
It is interesting to see the users prefer summary over other techniques.

**Reasons To Reject:**

The baselines for user study could be better - like just showing topics of sentences / or color each sentence based on their topics. Is it possible to run the study with more participants.

**Reproducibility:**

4: Could mostly reproduce the results, but there may be some variation because of sample variance or minor variations in their interpretation of the protocol or method.

**Reviewer Confidence:**

4: Quite sure. I tried to check the important points carefully. It's unlikely, though conceivable, that I missed something that should affect my ratings.

---

> ### Author Rebuttal · Authors · 2023-08-26
>
> Thank you for the valuable feedback. The idea of using sentence topics is certainly intriguing. However, with discussions spanning hundreds of sentences, predicting multiple topics might complicate the summary presentation. Evaluating the topic model on a sentence level without any ground truth adds to the complexity. Our search engine lets users explore their own queries without sifting through an extended list of topics. Nonetheless, we'll explore ways to integrate this suggestion practically in the future.
>
> 1. “__What happens if the discussion is single topic - where all users discuss the same topic ? Did authors find such discussions how did your approach tackle those__”
>
> In cases of single-topic discussions, we've noticed a dominant frame, yet it still contains around two or three clusters with related subtopics due to automatically identifying semantically related but slightly divergent sentence groups. For instance, for the topic "Corporations support the LGBT for money, not love," we observed three frames: "Economics" (241 sentences), "Fairness and Equality" (21 sentences), and "Public Opinion" (25 sentences). The "Economics" frame includes subtopics like "Corporations support LGBTQ+ for profit and image," "Corporations prioritize money over emotions," and more.
>
> 2. “__How much does the corpus based filtering important - an ablation study would have provided more details about this filtering step as this makes the approach corpus dependent__”
>
> Corpus-based filtering actually stands as a valuable component of our approach. It enables the use of a small noise sample per corpus/domain/platform to filter numerous emerging discussions. This process is made easier by the preliminary clustering, which easily groups meta comments due to their semantic similarity, aiding straightforward noise sampling. Unlike a generic preprocessing plan with several heuristics, our approach easily addresses distinct platform styles and content nuances, such as slang/terminology variations and content moderation policies (e.g., Reddit vs Discord vs Debate portals). Moreover, crafting rules for exclusion manually might prove ineffective without semantic similarity detection. We believe our clustering-based filtering offers a simpler, customizable solution adaptable to any corpus.
>
> 3. “__While user study evaluates whole discussion the offline evaluation is only based on clusters ? Is it possible to rather select N discussions and clusters in them to understand how the performance works - this will provide a better understanding like the quality of clusters formed, with-in cluster accuracy and across cluster accuracy of a method__”
>
> We actually did manually inspect 50 to 80 random clusters (with 50 to 100 sentences) from multiple discussions and were very satisfied with the cluster quality. This is also (albeit indirectly) reflected in the generated cluster labels as coherent inputs (good clusters) resulted in high quality cluster labels. Appendix B presents more details on the clustering setup. We acknowledge that this could be quantified better and will do so in future.
>
> Also, as we employed only one clustering algorithm (HDBSCAN), we didn’t utilize any automatic metrics (such as Dunn Index or Sum of Squared Distances) to evaluate cluster quality since these (usually) make sense only when compared to other clustering algorithms. We actually experimented with these but could not meaningfully interpret the quality just by the scores. Moreover, as we did not have ground truth cluster labels beforehand, computing accuracy would be impossible (e.g. using V-measure or Fowlkes-Mallows Index).

---

### Meta-Review · Area_Chair_Qdep · 2023-09-28

**Recommendation:** 4

**Metareview:**

The paper presents an unsupervised method, based on prompting and LLM, for making "indicative summaries" from forum discussions. Sentences are clustered and cluster labels are defined, which serve as a guide for reading the forum. Different language models are tested and an evaluation with human annotators is made.

---

### Decision · Program_Chairs · 2023-10-07

**Decision:**

Accept-Main

**Comment:**

The paper presents an unsupervised method, based on prompting and LLM, for making "indicative summaries" from forum discussions. Sentences are clustered and cluster labels are defined, which serve as a guide for reading the forum. Different language models are tested and an evaluation with human annotators is made.